# Is There a Better Source Distribution than Gaussian? Exploring Source Distributions for Image Flow Matching

**Junho Lee**[*]                                                                 *joon2003@snu.ac.kr*
*Graduate School of Data Science*
*Seoul National University*

**Kwanseok Kim**[*]                                                              *kjvd1009@snu.ac.kr*
*Graduate School of Data Science*
*Seoul National University*

**Joonseok Lee**[†]                                                             *joonseok@snu.ac.kr*
*Graduate School of Data Science*
*Seoul National University*

**Reviewed on OpenReview:** *https://openreview.net/forum?id=sev0GtV1fc*

## Abstract

Flow matching has emerged as a powerful generative modeling approach with flexible choices of source distribution. While Gaussian distributions are commonly used, the potential for better alternatives in high-dimensional data generation remains largely unexplored. In this paper, we propose a novel 2D simulation that captures high-dimensional geometric properties in an interpretable 2D setting, enabling us to analyze the learning dynamics of flow matching during training. Based on this analysis, we derive several key insights about flow matching behavior: (1) density approximation can paradoxically degrade performance due to mode discrepancy, (2) directional alignment suffers from path entanglement when overly concentrated, (3) Gaussian's omnidirectional coverage ensures robust learning, and (4) norm misalignment incurs substantial learning costs. Building on these insights, we propose a practical framework that combines norm-aligned training with directionally-pruned sampling. This approach maintains the robust omnidirectional supervision essential for stable flow learning, while eliminating initializations in data-sparse regions during inference. Importantly, our pruning strategy can be applied to any flow matching model trained with a Gaussian source, providing immediate performance gains without the need for retraining. Empirical evaluations demonstrate consistent improvements in both generation quality and sampling efficiency. Our findings provide practical insights and guidelines for source distribution design and introduce a readily applicable technique for improving existing flow matching models. Our code is available at https://github.com/kwanseokk/SourceFM.

## 1 Introduction

Flow Matching (FM) (Lipman et al., 2023; Liu et al., 2023b; Albergo et al., 2023) is a recently introduced approach for generative modeling that bridges ideas from Continuous Normalizing Flows (CNFs) (Chen et al., 2018; Grathwohl et al., 2018) and diffusion models (Sohl-Dickstein et al., 2015; Song & Ermon, 2019; Ho et al., 2020; Nichol & Dhariwal, 2021; Song et al., 2021b). The core idea involves training a vector field so that trajectories, following an ordinary differential equation (ODE), transport samples from a simple base distribution to a complex target distribution, offering advantages in stable training, fast inference, and explicit likelihood estimation. Unlike diffusion models, which typically involve a fixed stochastic process that

---

[*]Equal contribution.
[†]Corresponding author.

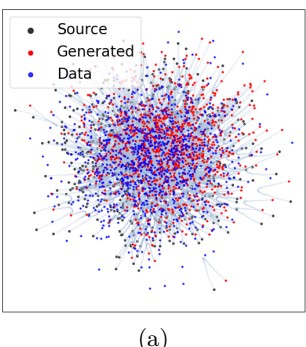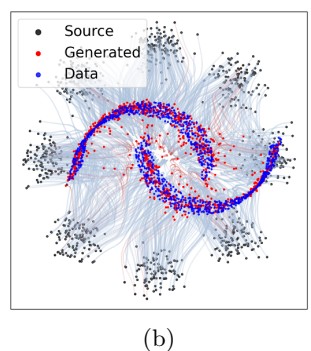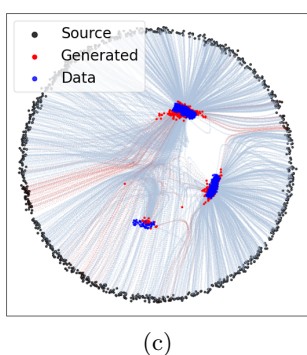

(a)           (b)           (c)

Figure 1: **2D simulations of flow matching.** The first two are examples of naive low-dimensional flow matching illustrations of (a) Gaussian to Gaussian and (b) 8 Gaussians to moons. Our proposed 2D simulation for flow matching is demonstrated in (c). Dots represent source (black), target data (blue), and generated samples (red). Blue dashed lines show successful ODE trajectories from source to generated samples, while red dashed lines indicate failed trajectories.

gradually adds Gaussian noise to data and requires learning a reverse denoising trajectory, flow matching offers complete freedom in defining the interpolation path between distributions. This flexibility not only enables the design of more efficient and straighter trajectories but also removes any constraint on the choice of source distribution.

This flexibility has motivated extensive research on enhancing FM performance through improved source–target pairing strategies (Tong et al., 2024; Pooladian et al., 2023; Davtyan et al., 2025) and developing straighter probability paths (Xing et al., 2023; Lee et al., 2023; Liu et al., 2023b) to reduce trajectory curvature and minimize function evaluations. Recent advances have leveraged prior knowledge of target distributions to construct better-aligned source distributions (Stark et al., 2023; Jing et al., 2024; Kollovieh et al., 2025), demonstrating substantial improvements through partial target information incorporation. For multimodal generation, recent works have exploited this source flexibility by using text embeddings directly as source distributions (Liu et al., 2024; He et al., 2025).

However, most existing target-aware source designs are still limited to relatively simple or low-dimensional datasets, such as time series (Kollovieh et al., 2025) or protein-ligand interactions (Stark et al., 2023). To the best of our knowledge, there has been little to no prior work applying them to complex, high-dimensional data like natural images—likely due to the challenging nature of such applications. Moreover, naively replacing the standard Gaussian source with distributions from other modalities (*e.g.*, text embeddings) often fails to achieve stable target mappings, requiring extensive engineering as shown by prior work (Liu et al., 2024; He et al., 2025). This raises a fundamental question: What makes deviating the source from the Gaussian distribution so difficult, and what properties are required for a source to be advantageous in flow matching? To answer this, we begin with an intuitive hypothesis: source distributions that more closely approximate the density or geometric characteristics of the target distribution would lead to improved performance and faster convergence.

Since learning behavior in high-dimensional datasets is difficult to interpret and visualize, we begin with low-dimensional experiments that enable clear visual analysis of the underlying learning dynamics. Several prior works have used naive low-dimensional experiments to demonstrate their effectiveness, *e.g.*, 2D Gaussian to 2D Gaussian in Fig. 1(a) or 8 Gaussians to moons in Fig. 1(b). However, we discover that merely using a low-dimensionality does not reflect the intricate but essential learning dynamics that emerge only in high-dimensional cases, and thus lessons learned from such a naive low-dimensional toy experiment do not fully transfer to the actual applications of interest, which typically involve high-dimensional data. To address this, we redesign the low-dimensional experiments in Section 3 to better reflect the geometric properties of high-dimensional data by decomposing it into direction and norm components, as illustrated in Fig. 1(c). This 2D simulation allows us to more realistically analyze the learning dynamics of flow matching, preserving

the essential characteristics of high-dimensional applications while enabling clear visualization of vector field behaviors in 2D.

Using this novel experimental framework, we examine in Section 4 how different source distribution strategies interact with various pairing methods (independent *vs.* in-batch optimal transport). This allows us to test our intuitive hypothesis that source distributions more closely approximating the density or geometric characteristics of the target distribution should improve performance. Specifically, we propose three strategies to redesign the source distribution: progressively approximating the density of the target distribution (*density approximation*), designing sources based on directional information of the target distribution (*directional alignment*), and aligning the expected norms of the source and target distributions (*norm alignment*).

From our simulation results and analysis in Sections 4 and 5, we derive several critical insights that reshape our understanding of source distribution design in flow matching: (1) Source distributions closely approximating the target density can paradoxically degrade performance, with stronger approximations leading to worse outcomes—a phenomenon we attribute to mode discrepancy where approximate sources omit low-density data modes. (2) Directional alignment strategies, while theoretically appealing, often suffer from path entanglement due to insufficient source support and imperfect pairing, both of which hinder optimal performance. (3) The standard Gaussian source with independent pairing demonstrates unexpected robustness through omnidirectional coverage, while in-batch optimal transport pairing, despite its efficiency, sacrifices this crucial omnidirectional learning. (4) Regions with sparse or missing data consistently lead to generation failures due to insufficient supervision of the vector field during training. (5) A significant discrepancy between the norm distributions of the source and target incurs a substantial learning cost to resolve. These patterns hold consistently across both our controlled 2D simulations and real-world high-dimensional image datasets, demonstrating that our framework captures fundamental dynamics of flow matching.

Our key findings suggest that an effective source distribution design should leverage specific advantages while mitigating known weaknesses. Accordingly, we propose a new hybrid framework that combines training on a Gaussian source with "Norm Alignment" and employing "Pruned Sampling" during inference. This approach preserves the benefits of robust, omnidirectional learning from the Gaussian source (Finding 3) while resolving the learning inefficiency caused by norm mismatch (Finding 5). Furthermore, Pruned Sampling directly tackles the issue of passing through regions where the vector field is inaccurately learned (Finding 4) by excluding sampling from data-sparse regions, thus improving the quality of the final output. Notably, Pruned Sampling can be applied post-hoc to improve the performance of existing, pre-trained models without any need for retraining. This provides a practical pathway to immediately upgrade a wide range of flow matching models.

To summarize, we first propose a novel set of 2D simulation experiments designed to reveal and interpret the complex learning dynamics of flow matching in high-dimensional settings. Through these experiments, we derive critical findings that lead to a set of practical guidelines for source distribution design. Finally, we propose a readily applicable pruned sampling technique that can significantly improve existing flow matching models without the need for retraining. We believe these contributions not only advance the understanding of the flow matching field but also suggest new directions for developing more efficient and robust generative models.

## 2 Background and Related Work

### 2.1 Flow Matching

Continuous Normalizing Flows (CNFs) (Chen et al., 2018; Grathwohl et al., 2018) define an ODE $dx = u_t(x)dt$ with a time-varying velocity field $u : [0, 1] \times \mathbb{R}^d \to \mathbb{R}^d$ that transports an initial distribution $q_0$ to a target distribution $q_1$ over $t \in [0, 1]$. This ODE induces a path of densities $p_t(x)$ satisfying the continuity equation $\partial_t p_t(x) + \nabla \cdot \big(p_t(x) \, u_t(x)\big) = 0$. Flow Matching (FM) (Albergo et al., 2023; Lipman et al., 2023; Liu et al., 2023b) is a simulation-free approach for training CNFs that avoids trajectory simulation by regressing the model's vector field to a known probability flow. Instead of maximizing log-likelihood, FM assumes a prescribed probability path $p_t(x)$ that smoothly interpolates from $p_0 = q_0$ to $p_1 = q_1$, along with its

corresponding velocity field $u_t(x)$. The neural ODE $v_\theta(x, t)$ is trained to match $u_t(x)$ via

$$\mathcal{L}_{\text{FM}}(\theta) = \mathbb{E}_{t \sim \mathcal{U}[0,1], x \sim p_t(x)} \left[ \left\| v_\theta(x, t) - u_t(x) \right\|^2 \right]. \tag{1}$$

FM is compatible with various interpolation paths (*e.g.*, linear or Gaussian) and enables scaling CNFs to high-dimensional data while maintaining stable optimization. However, this objective is often intractable due to the need to evaluate the marginal $p_t(x)$ and vector field $u_t(x)$. Conditional Flow Matching (CFM) addresses this by introducing a latent condition $z$ to index a family of probability paths. Instead of a single fixed $p_t$, CFM considers a conditional density $p_t(x \mid z)$ with associated vector field $u_t(x|z)$. The overall path is recovered as $p_t(x) = \mathbb{E}_{z \sim q(z)}[p_t(x|z)]$. The CFM objective is given by

$$\mathcal{L}_{\text{CFM}}(\theta) = \mathbb{E}_{t \sim \mathcal{U}[0,1], z \sim q(z), x \sim p_t(x|z)} \left[ \left\| v_\theta(x, t) - u_t(x|z) \right\|^2 \right]. \tag{2}$$

As it satisfies $\nabla_\theta \mathcal{L}_{\text{CFM}}(\theta) = \nabla_\theta \mathcal{L}_{\text{FM}}(\theta)$ under mild conditions, optimizing $\mathcal{L}_{\text{CFM}}$ learns the correct global flow while sampling only from simpler conditional distributions. The key practical insight of CFM is to define the latent variable $z$ using the data samples themselves. In the standard independent coupling scheme, one sets $z = (x_0, x_1)$, where $x_0 \sim p_0(x)$ is a sample from the source distribution and $x_1 \sim p_1(x)$ is a sample from the target. The conditional path $p_t(x \mid x_0, x_1)$ thus becomes a simple "bridge" between a single source point and a single target point, making its path and vector field trivial to compute. Consequently, the training process only requires the ability to sample from $p_0(x)$ and $p_1(x)$, removing any need to evaluate its probability density (or log-density). This provides immense flexibility in the choice of $p_0$.

With this simple framework, FM is able to model more flexible probability paths (Chen & Lipman, 2024; Gat et al., 2024; Stark et al., 2024; Cheng et al., 2025; Kapusniak et al., 2024) than diffusion models (Song et al., 2021a; Song & Ermon, 2019; Song et al., 2021b; Rombach et al., 2022; Dhariwal & Nichol, 2021). FM has also been extended to various domains, including image (Esser et al., 2024; Dao et al., 2023; Ren et al., 2024), audio (Guan et al., 2024; Liu et al., 2023a; Prajwal et al., 2024), video (Jin et al., 2025; Polyak et al., 2024), molecule (Dunn & Koes, 2024; Song et al., 2023), and text generation (Hu et al., 2024).

## 2.2 Optimal Transport CFM (OT-CFM)

When trained with a globally optimal coupling, the flow map learned by FM aligns with the $\mathcal{W}_2$ geodesic between the source and target distributions. However, in practice, CFM typically adopts a simpler independent coupling $q_{\text{ind}}(x_0, x_1) = p_0(x_0) p_1(x_1)$, which pairs each source sample $x_0 \sim p_0$ with a randomly chosen target sample $x_1 \sim p_1$ (I-CFM). Given any coupling $q \in \Pi(p_0, p_1)$, where $\Pi(p_0, p_1)$ denotes the set of all valid joint distributions with marginals $p_0$ and $p_1$, the transport cost and the optimal transport plan $\pi$ are defined as:

$$\mathcal{C}(q) = \mathbb{E}_{(x_0, x_1) \sim q} \left[ \|x_1 - x_0\|^2 \right], \qquad \pi = \underset{q \in \Pi(p_0, p_1)}{\arg\min} \ \mathcal{C}(q). \tag{3}$$

The independent coupling $q_{\text{ind}} = p_0(x_0)p_1(x_1)$ generally incurs a suboptimal transport cost. In this paper, we define the excessive cost relative to the optimal transport plan $\pi$ as the *Wasserstein coupling gap*:

$$\Delta_{\text{W}} = \mathcal{C}(q_{\text{ind}}) - \mathcal{C}(\pi) \ \geq \ 0. \tag{4}$$

This gap quantifies the degree of misalignment between the randomly paired paths and the true geodesic paths. A large gap forces the learned flow to exhibit unnecessary curvature, deviating from the straightest possible transport map, which can in turn increase gradient variance and hinder training efficiency. To mitigate this issue, mini-batch optimal transport (BatchOT or OT-CFM) (Pooladian et al., 2023; Tong et al., 2024) selects, at each training step, the permutation that best matches the $B$ source samples to the $B$ target samples, yielding a locally optimal Wasserstein coupling. Because this permutation is recomputed for each mini-batch, the alignment does not persist across iterations, which can limit long-term convergence benefits.

## 2.3 Alternative Source Distributions

**Target-Approximating Source Distribution.** TSFlow (Kollovieh et al., 2025) extends target-approximating source distributions to time series forecasting, integrating Gaussian Process priors within CFM

to align source distribution with temporal target data structure. HarmonicFlow (Stark et al., 2023) employs physics-based priors for protein-ligand docking through 3D geometric constraints, utilizing protein backbone torsion-angle distributions to preserve SE(3)-equivariant properties. AlphaFlow (Jing et al., 2024) learns directly from PDB structural databases, replacing Gaussian noise with experimentally observed distributions to model biomolecular conformational variability. While promising, these approaches remain limited to simple domain-specific distributions and low-dimensional spaces.

**Non-Gaussian Source in Cross-Modal Generation.** CrossFlow (Liu et al., 2024) enables direct mapping from text to image embeddings using discrete language spaces as source distributions, overcoming traditional Gaussian space constraints. FlowTok (He et al., 2025) constructs a shared text-image token space by transforming Vision Transformer patch tokens into quantized discrete spaces, redefining image generation as 1D token sequence generation. However, these approaches require substantial engineering for source-target alignment, suggesting naive replacements may create more problems than solutions for complex high-dimensional data.

### 2.4 Von Mises-Fisher (vMF) Distribution

The von Mises-Fisher (vMF) distribution (Fisher, 1953), denoted as $\text{vMF}(\mu, \kappa)$, provides a mathematically principled framework for this directional concentration, defining a probability distribution on the unit hypersphere with mean direction $\mu$ and concentration parameter $\kappa$. As $\kappa$ increases, the distribution becomes more sharply concentrated around the mean direction $\mu$. When $\kappa = 0$, the distribution reduces to a uniform distribution over the unit sphere. This makes the vMF distribution particularly suitable for modeling directional data or sampling unit vectors with controlled angular concentration, which is useful in flow matching tasks where aligning the directionality of source and target samples is crucial.

## 3 Revisiting Gaussian Source Distributions for Flow Matching

The choice of the source distribution $p_0$ is a fundamental element in flow matching (Lipman et al., 2023; Liu et al., 2023b; Tong et al., 2024). While the standard isotropic Gaussian distribution $\mathcal{N}(0, I)$ is widely adopted for its simplicity and favorable mathematical properties, its optimality for complex image generation tasks remains as an open question. In this section, we point out geometric properties of Gaussian distributions, propose a functionally equivalent directional sampling paradigm, and derive key insights for designing effective source distributions with 2D simulations.

### 3.1 High-Dimensional Geometry and Directional Decomposition

The standard Gaussian $x_G \sim \mathcal{N}(0, I) \in \mathbb{R}^d$ serves as a conventional choice for the source distribution in generative models. In high-dimensional spaces, the vast majority of independent and identically distributed (i.i.d.) Gaussian samples reside on a thin *hyperspherical shell*. Specifically, the Euclidean norm of $x_G$ follows $\chi$-distribution: $\|x_G\|_2 \sim \chi(d)$, whose mean and variance are approximately $\sqrt{d - 1/2}$ and $1/2$, respectively. This indicates that for a sufficiently large $d$, most samples lie on a hypersphere whose radius is sharply concentrated in a narrow band around $\sqrt{d - 1/2}$. Motivated by this property, we propose the $\chi$-Sphere decomposition, a directional decomposition scheme defined as:

$$x_G \approx r s_G, \quad \text{with } r \sim \chi(d),\ s_G \sim \mathcal{U}(S^{d-1}), \tag{5}$$

where $\mathcal{U}(S^{n-1})$ denotes the uniform distribution on the unit sphere. Our $\chi$-Sphere decomposition preserves all Gaussian statistics while explicitly factorizing each sample into an independent radius and a directional unit vector (see the derivation in Section A). To verify that this theoretical equivalence holds in practice, we further measure the FID scores of generated images using different combinations of training and inference source distributions. The result in Table 1 indicates that the samples from $\chi$-Sphere and standard Gaussian distributions are interchangeable during both training and inference, without showing significant performance difference. Consequently, we regard these two sampling methods as functionally equivalent throughout this paper.

In practice, normalized data–typically shifted to zero mean and scaled to lie within $[-1, 1]$–tend to concentrate within a bounded norm region, forming a thick *hyperspherical shell*. This enables a similar decomposition of data point $x_1 \equiv r_1 s_1 \in \mathbb{R}^d$, where $s_1 \in \mathbb{R}^d$ is the unit vector representing its direction and $r_1 \in \mathbb{R}$ is the norm of the data sample. Under this formulation, we can reasonably measure the "angular similarity" between a sample from Gaussian and a data point using the cosine similarity, $s_G \cdot s_1$. Viewing the data distribution through this directional perspective reveals it as a subset of directions of a full hypersphere, a viewpoint we explore further in Section 4.2.

## 3.2 Proposed Simulation for High-Dimensional Flow Analysis

While previous works have visualized flow matching between simple 2D distributions, there has been little attention to understanding the learning dynamics of flow models in high-dimensional settings. Davtyan et al. (2025), for example, illustrates 2D Gaussian to 2D Gaussian, shown in Fig. 1(a). Tong et al. (2024), illustrated in Fig. 1(b), visualizes the transport paths from the 8 Gaussians to moons. These settings, however, fail to reflect the geometric properties of source and target samples and do not capture the behavior of flow matching in high-dimensional settings. To address this gap and analyze the behavior of flow matching models in high-dimensional settings, we propose a novel flow simulation setup that is more suitable for capturing the geometric properties uniquely observed in high-dimensional distributions.

Specifically, we simulate high-dimensional geometric structure in 2D by sampling random directions $\theta \in [0, 2\pi]$ and scaling with norms from $\chi(d)$ distributions (● Source) following the $\chi$-Sphere decomposition, as illustrated in Fig. 1(c). When analyzing flow matching with the proposed density-approximated sources and directional sources introduced in Section 1, we normalize their mean norms to match the $\chi$-Sphere radius. This normalization allows us to observe transport path patterns comparable to the $\chi$-Sphere while ensuring similar geometric properties across all source types. For the target data (● Data), we sample points along three clusters with varying densities, reflecting the common scenario where real high-dimensional datasets scaled to $[-1, 1]$ (*e.g.*, CIFAR-10 with norm 27.2) typically have smaller norms than their corresponding Gaussian sources (55.4) and reside inside the concentration shell. We train flow matching models on these geometrically designed distributions and visualize their transport trajectories. Blue dashed lines (-- ODE trajectory) show successful paths from source (● Source) to generated samples (● Generated), while red dashed lines (-- Bad ODE trajectory) indicate failures. Here, we define success/failure based on whether the L2 distance from generated samples to the nearest data point is within one unit.

To complement our primary visual analysis, we report the Normalized Wasserstein (Balaji et al., 2019), averaged over 10 runs, as an auxiliary measure. This metric captures how closely the generated samples resemble the true data distribution, even when the data contains multiple modes with imbalanced proportions. Unlike Wasserstein distance, the Normalized Wasserstein is robust to differences in mode proportions and focuses on the structural similarity between distributions. Lower values indicate better alignment with the true distribution. Since our setup is a controlled simulation, this metric should not be interpreted in absolute terms. Instead, it is most meaningful when used for relative comparison within the same group of source distributions. Additional metrics and implementation details are provided in Section B.

## 4 Understanding Learning Dynamics of FM: Insights from 2D Simulations

Using the simulation approach proposed in Section 3.2, we conduct comprehensive experiments to verify the source distribution strategies proposed in Section 1. Although simplified, these experiments preserve the key geometric structures and transport dynamics of high-dimensional settings. They allow us to empirically

Table 1: FID scores under different combinations of training and inference source distributions.

| Training \ Inference | Gaussian | $\chi$-Sphere |
|---|---|---|
| Gaussian | 4.40 | 4.29 |
| $\chi$-Sphere | 4.45 | 4.42 |

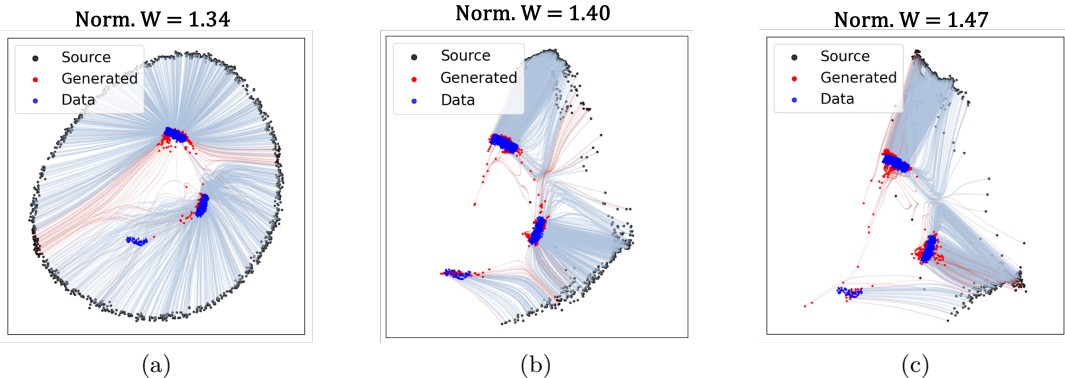

Figure 2: **Visualization of flow matching with density-approximated source.** (a) OT-CFM with Approximated Source (iter 200), (b) OT-CFM with Approximated Source (iter 6000), (c) OT-CFM with Approximated Source (iter 10000). "Norm. W" denotes normalized Wasserstein, where a lower value indicates better generation performance.

probe the core mechanisms behind flow matching, offering insights into what drives its success or failure across different conditions.

## 4.1 Density Approximation Strategy

Recent studies (Pooladian et al., 2023; Tong et al., 2024) have shown that improving the pairing between source and target distributions produces straighter probability flow trajectories, fewer sampling steps at inference, and expedited training. These improvements are largely attributed to reduced transport costs achieved through better alignment. Motivated by these findings, we hypothesize that shaping the source distribution to closely resemble the target distribution—while preserving the improved pairing—might further reduce transport cost and improve generation performance.

To test our hypothesis, we first train a flow matching model, which we refer to as the *density approximator*, to transport samples from the $\chi$-Sphere Gaussian distribution toward the target data distribution. The density approximator is trained for different numbers of iterations (specifically, 200, 6,000, and 10,000) to achieve progressively stronger approximations of the target distribution, and at each stage, we use the samples generated by the partially trained model as the new approximated source distribution, denoted by $\tilde{p}_0$. To ensure a consistent visual comparison across these stages, we then scale each $\tilde{p}_0$ to match the average norm of the original $\chi$-Sphere, as interpreting the transport dynamics without this alignment is challenging, as illustrated in Fig. 8. This approach allows us to observe how the progressively approximated source distribution $\tilde{p}_0$ acts as a new source distribution for flow matching.

Contrary to our initial hypothesis that a source distribution resembling the target would improve generation performance, we observe in Fig. 2 that more samples fall outside the data modes as the source approximates the target data. Also, higher Normalized Wasserstein is observed when the source distribution becomes increasingly similar to the target distribution. This indicates that the performance degrades in spite of a better density approximation between the source and the target, implying that a more similar source distribution might hinder accurate mode coverage and generation quality. This counterintuitive outcome challenges our hypothesis about source distribution design in flow matching and necessitates careful analysis of the underlying mechanism.

**Mode Discrepancy.** Taking a deeper look, as the source density approximator gradually learns to match the target distribution, the approximated source $\tilde{p}_0$ concentrates samples around each data mode (● Data). Subsequently, the distribution is adjusted to spread outward (● Source), aligning its norm with that of the $\chi$-Sphere source, as discussed in Section 3.2. Comparing Fig. 2(a), (b) and (c), the density approximator reflects the two dominant data modes while gradually departing from the general spherical shell structure of the $\chi$-Sphere. However, $\tilde{p}_0$ fails to capture the lower-left mode in sparse density regions even after 10,000

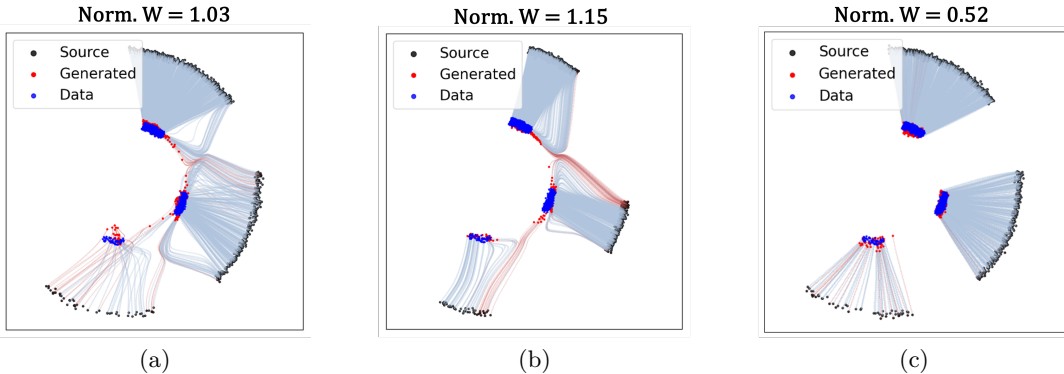

Figure 3: **Visualization of flow matching with ideally aligned directional source.** (a) OT-CFM with directional source, (b) OT-CFM with tight directional source, (c) global OT Pairing with directional source. 'Norm. W' denotes Normalized Wasserstein.

training iterations. These 2D simulation results reveal a fundamental problem that occurs in practice: any approximated distribution $\tilde{p}_0$ inevitably suffers from information loss, particularly in low-density regions where several modes may be underrepresented or omitted. This inaccurate approximation creates a pairing mismatch where certain target data points $x_1$ from underrepresented or omitted modes cannot be effectively paired with appropriate source samples $x_0$ from the approximated source $\tilde{p}_0$. We term this phenomenon *mode discrepancy.* Consequently, even optimal transport pairing is forced to learn inefficient and complex trajectories, potentially increasing the learning difficulty instead of reducing it. This analysis reveals that mode discrepancy poses a fundamental limitation to density approximation strategies for flow matching source distributions.

## 4.2 Directional Alignment Strategy

Recognizing the critical importance of preserving all data modes during approximation, we shift our focus toward a different approach that sacrifices norm information to better approximate data mode directions. Specifically, we leverage $\chi$-Sphere decomposition and focus on the cosine similarity between the source unit vector $s_G$ and the target unit vector $s_1$. In an ideal scenario where we know all data mode directions, we can incorporate this knowledge into our source modeling. For each data point belonging to a specific mode direction, we construct a $\chi$-Sphere source that maintains a cosine similarity above a threshold $\gamma$ with the data (*i.e.*, $s_G \cdot s_1 \geq \gamma$). This approach allows us to avoid the mode discrepancy problem encountered earlier. This directional alignment approach is visualized under our experimental setting in Fig. 3, where the source successfully reflects all data mode directions. Combined with the improved pairing strategy, we expect this configuration to yield optimal results.

**Imperfect Pairing.** However, even when using a source distribution that covers all data modes through directional alignment and employing mini-batch OT pairing method, the configuration fails to achieve the expected optimal performance. As shown in Fig. 3(a) and (b), we observe that points are not generated perfectly within each data cluster—some red points scatter outside the blue target clusters. This suboptimal behavior contrasts with the global OT-paired directional alignment source in Fig. 3(c), where all generated points successfully converge to their respective data clusters. These findings highlight the limitation of mini-batch OT pairing that still falls significantly short of perfect global OT pairing.

**Path Entanglement.** In addition, interestingly, we observe that using a more tightly-aligned directional source by increasing $\gamma$ in the directional source alignment actually leads to performance degradation, as shown in Fig. 3(b). From a geometric similarity perspective, as $\gamma$ approaches 1, Fig. 3(b) shows that the source distribution becomes increasingly concentrated, spreading only as much as the data itself. However, we find that this reduction in source support adversely affects the overall performance. The underlying cause of this phenomenon can be seen in Fig. 3(b), where paths from each source to target cluster become heavily entangled

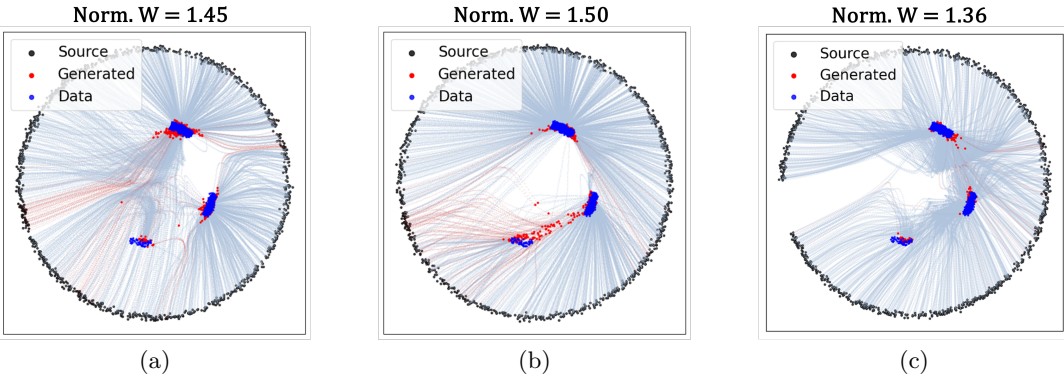

Figure 4: **Visualization of flow matching with different methods.** (a) I-CFM with Gaussian source (b) OT-CFM with Gaussian source (c) I-CFM with pruned source.

in localized regions during generation. This excessive entanglement leads to highly inconsistent vector field directions within local areas, making the vector field difficult to learn and unstable during optimization.

To better understand the cause of this entanglement, we provide a mathematical analysis of how the geometry of source-target pairings changes under increased concentration of the source distribution. Consider two target points $x_1, x_1'$ within the same mode cluster, with their associated source points $x_0, x_0'$ drawn from the corresponding directional source cluster. Their linearly interpolated paths are given by $x_t = (1-t)x_0 + tx_1$ and $x_t' = (1-t)x_0' + tx_1'$, respectively, with separation: $d(t) = x_t - x_t' = (1-t)(x_0 - x_0') + t(x_1 - x_1')$. This separation obeys $\min_{t \in [0,1]} \|d(t)\| \leq \|x_0 - x_0'\| = \mathcal{O}((1-\gamma)^{1/2})$. Therefore, large values of $\gamma$ cause the trajectories to be nearly coincident at initialization, which in turn leads to a sharp increase in the required local Lipschitz constant:

$$L_{\text{local}} \gtrsim \frac{\sin\theta}{\|x_1 - x_1'\| + \mathcal{O}((1-\gamma)^{1/2})}, \quad \text{where} \quad \theta = \arccos\left(\frac{(x_1 - x_0)^\top (x_1' - x_0')}{\|x_1 - x_0\| \, \|x_1' - x_0'\|}\right). \tag{6}$$

As a result, $L_{\text{local}}$ becomes larger, making the optimization increasingly unstable and slow. More detailed derivation is provided in Section C.

These findings demonstrate that without oracle-level pairing, learning becomes significantly more difficult due to inconsistent vector fields induced by trajectory interference. To avoid this, the source distribution must retain the angular support sufficiently. The limitations observed here also extend to the density approximation discussed in Section 4.1, where increased approximation leads to reduced support, and imperfect pairing continues to pose challenges.

In conclusion, contrary to our initial expectation that combining mini-batch OT with improved source approximation (density or direction) would yield optimal results, our simulations reveal that increased approximation actually leads to diminished performance due to mode discrepancy, imperfect pairing mechanisms, and path entanglement. This counterintuitive finding underscores the delicate balance required between source distribution design and coupling strategies in flow matching approaches.

### 4.3 Pairing Method Analysis

To understand how different pairing methods affect the learning dynamics of flow models, we further analyze the generation trajectories and learning patterns of each approach.

**Independent Pairing (I-CFM).** The generation trajectory of I-CFM with $\chi$-Sphere Gaussian source in Fig. 4(a) reveals patterns that can be interpreted as a conceptually two-stage transport process. In the initial stage, samples from the source move coarsely toward data-dense regions where the data modes are located. After this rough transport, fine-grained adjustments are performed in the second stage, where the paths are converged from all directions towards each mode. This occurs because the $\chi$-Sphere source distribution provides uniform coverage across all directions. When paired independently with the target data, each data

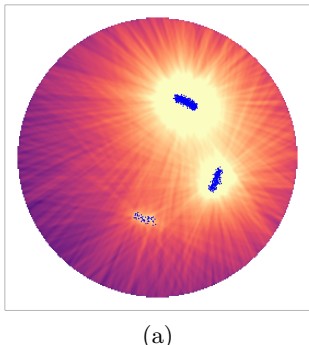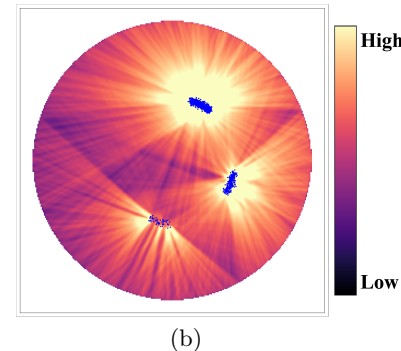

(a)                                                    (b)

Figure 5: **Visualization of flow trajectory heatmap.** These heatmaps show the distribution of paths learned by each model. Color intensity corresponds to trajectory density, where brighter regions indicate higher concentrations of paths. (a) I-CFM and (b) OT-CFM.

sample receives supervision from sources distributed omnidirectionally, enabling comprehensive vector field learning around each mode. This is clearly observed in Fig. 5(a), which shows the density of trajectories actually learned during training with independent pairing. The same principle applies to high-dimensional Gaussians, which also provide uniform directional coverage.

**Improved Pairing (OT-CFM).** In contrast, the OT-CFM trajectory heatmap in Fig. 5(b) shows a markedly different pattern. OT-CFM uses a batch-level optimal transport pairing strategy that minimizes the transport cost within each batch. This in-batch optimal pairing creates a more localized pairing where most source samples are matched with the nearby target cluster. Consequently, the model learns denser and more linear trajectories between optimally paired source and target samples. This can be seen in Fig. 4(b), where many source samples follow straighter paths compared to the more curved trajectories in Fig. 4(a) produced by I-CFM.

However, this improved path efficiency comes with a drawback. Because OT-CFM consistently pairs samples with nearby targets, the model learns the vector field primarily along narrow, cone-shaped directions centered around each data mode. As a result, it fails to learn the omnidirectional vector fields near the modes, which I-CFM captures better. This leads to insufficient learning of vector fields in broad angular directions, especially from the origin to the data modes, as shown in the trajectory heatmap in Fig. 5(b).

If all pairings were perfect, this focused learning approach would be both efficient and effective, as the model could learn the vector field accurately and consistently along these specific trajectories. In practice, however, the pairing is inevitably imperfect. When samples need to traverse these undertrained regions due to suboptimal pairings, they lack proper vector field guidance. Consequently, these samples often stall or veer off course, failing to reach the target data. This failure is clearly illustrated in Fig. 4(b), where multiple red points appear stranded midway, outside the data modes. This reveals a fundamental trade-off in OT-CFM: improved path efficiency comes at the expense of the robustness provided by omnidirectional coverage.

**Low-density Directions.** A common observation in both Fig. 4(a) and (b) is that when samples are initialized in *low-density directions*–directions where data is absent and the angular separation from data modes is large–they often fail to reach the data manifold during generation. These failed trajectories appear as clusters of red dashed lines ( -- ) near this direction. This issue arises because, interpolation paths from directions with sparse or absent data are sampled relatively less frequently during training, leading to lower training frequency for these regions. This is clearly visible in Fig. 5, where darker regions (such as the left and lower left areas) indicate lower training frequency compared to other parts of the heatmap. The rarity of path traversal during training results in insufficient supervision for the velocity field in these regions. Consequently, the vector field in such areas becomes inaccurately learned, increasing the likelihood of generation failures from samples initialized in low-density directions.

Building upon these insights, we experiment with a simple strategy that reflects the flow learning dynamics observed in our simulation. By comparing the two pairing methods, we find that learning an omnidirectional vector field around each data mode leads to a more robust velocity field. We also observed that, across all pairing strategies, source samples from low-density directions tend to produce poor generations. Combining

Table 2: **Comparison of Approximated Target Distribution.** Unconditional generation results on CIFAR10 using different source distributions. All models are trained with OT-CFM (Tong et al., 2024).

| | Gaussian | DCT-Weak | DCT-Strong | GMM-1 | GMM-2 | GMM-10 | CNF |
|---|---|---|---|---|---|---|---|
| FID ↓ | 4.40 | **4.20** | 4.39 | 11.75 | 12.49 | 12.11 | 17.18 |

these findings, we train I-CFM using the full $\chi$-Sphere source to encourage robust and omnidirectional learning. Then, during sampling, we prune source samples from low-density directions to avoid poorly trained regions. This hybrid strategy, which is visualized in Fig. 4, results in significantly fewer failed trajectories and improved performance by maintaining comprehensive training coverage while guiding sampling toward more reliable paths during generation. Through this approach, we gained valuable insights that could help identify better source distributions for flow matching models.

## 5 Empirical Validation on High-Dimensional Image Datasets

In this section, we extend our analysis to high-dimensional image datasets through parallel experiments that validate the hypotheses and findings from Section 4. We investigate whether the insights from our low-dimensional simulations transfer to real-world image generation tasks.

### 5.1 Density Approximation Strategy

In Section 4.1, we examined the intuitive hypothesis that using source distributions that better approximate the target distribution's density would improve the efficiency and performance of flow matching through our 2D simulations. Our findings revealed that density-approximated sources introduce mode discrepancy issues that hinder effective training and yield inferior performance compared to baseline. To evaluate whether this conclusion in 2D simulation can be generalized to high-dimensional real-world datasets, we investigate three progressively expressive approximations to the target distribution and assess their impact on flow matching performance and training dynamics. To evaluate progressively stronger approximations to the target distribution, we employ the following density approximation methods:

- **Discrete Cosine Transform (DCT)** (Ahmed et al., 1974; Strang, 1999), commonly used in image compression (*e.g.*, JPEG (Wallace, 1992)), separates spatial frequencies of the image, discarding high frequency components that are less perceptible to humans (Wang et al., 2004). Inspired by this, we apply DCT filtering to Gaussian source to remove less prominent high-frequency components of the target distribution. See Section D for more details.
- **Gaussian Mixture Model (GMM)** (Reynolds et al., 2009) approximates a complex distribution by fitting multiple Gaussians. We train GMMs with 1, 2, and 10 components using the Expectation-Maximization (EM) algorithm (Moon, 1996), and use these as alternative source distributions.
- **Continuous Normalizing Flow (CNF)** (Grathwohl et al., 2018; Chen et al., 2018) constructs flexible distributions by continuously transforming a simple base density through a series of invertible mappings parameterized by neural ODEs. Although its invertibility constraint might limit expressiveness, CNF can still serve as a sophisticated approximation of the data distribution. We train FFJORD (Grathwohl et al., 2018), a popular CNF model, and use its output as the source distributions for flow matching.

As shown in Table 2, while DCT-based mild refinement improves performance compared to the Gaussian baseline, stronger approximations to the target distribution progressively degrade performance (DCT > GMM > CNF in terms of quality). This finding aligns with our simulation results from Section 4, where we observed similar performance degradation with stronger density approximations.

Although high-dimensional settings make it difficult to analyze the specific underlying mechanisms, the consistent trend across both simulations and real experiments suggests a fundamental issue with density-based approximation strategies. Drawing from our 2D analysis, a key insight is that approximating the target distribution inevitably leads to information loss, particularly in low-density regions. These regions tend

Table 3: **Directional Alignment Performance.** FID scores on CIFAR10 using Euler integration for 100 NFE, comparing standard Gaussian source with directionally-aligned alternatives.

| Algorithm | Oracle-vMF | | | Kmeans-vMF | | | | | Gaussian |
|---|---|---|---|---|---|---|---|---|---|
| | $\kappa = \infty$ | $\kappa = 3000$ | $\kappa = 1500$ | $\kappa = 1500$ | $\kappa = 900$ | $\kappa = 300$ | $\kappa = 100$ | $\kappa = 50$ | $(\kappa = 0)$ |
| OT-CFM | 0.74 | 1.98 | 2.86 | 7.11 | 5.86 | 4.64 | 4.22 | 4.15 | 4.40 |

to be underrepresented or omitted in the support of the approximate source $\tilde{p}_0$. As demonstrated in our 2D simulation (Fig. 2(b)), when no source sample $x_0 \sim \tilde{p}_0$ is directionally aligned with a given target sample $x_1 \sim p_1$ (such as samples from sparse modes), even OT-based coupling produces inefficient, entangled trajectories that increase training complexity and hurt generative performance.

## 5.2 Directional Alignment Strategy

In our previous analysis, we identified a fundamental limitation: approximating the target distribution's density inevitably leads to information loss, and using such approximations as source distributions results in mode discrepancy, where certain data modes lack corresponding suitable samples, hindering effective training. To address this challenge, we propose a directional alignment strategy that focuses on comprehensive mode coverage, paralleling our 2D simulations in Section 4.2.

We find that the FID between normalized and original data is only 0.29, demonstrating that directional information is the key factor for generation quality. Based on this insight, rather than attempting to match the complete target density—which suffers from information loss in sparse regions—we focus on preserving directions where target data exists while removing directions without data. This approach prioritizes directional information of data modes while ignoring norm information, thereby avoiding the mode discrepancy problems inherent in density approximation strategies.

To obtain a distribution aligned with target directions, we leverage the $\chi$-Sphere decomposition introduced in Section 3.1 and vMF distribution. This allows us to approximate directional pairing by generating source samples $x_0$ that are directionally aligned with each target sample $x_1$. The specific sampling strategy follows:

$$x_0 = r\, s_0, \quad \text{with } r \sim \chi(d), \, s_0 \sim \text{vMF}\big(\mu(x_1), \kappa(x_1)\big), \tag{7}$$

where the vMF's mean direction $\mu(x_1)$ aligns the direction of $x_0$ with $x_1$, while the probabilistic radius $r$ and concentration parameter $\kappa(x_1)$ enable probabilistic rather than deterministic pairing, providing robust alignment through controlled variability.

### 5.2.1 Oracle Approach

To investigate the potential performance when sampling is focused on directions where data actually exists, we design an oracle pairing experiment. In this oracle scenario, the vMF mean direction is set to each normalized data point, and varying $\kappa$ values control the tightness of pairing between source and target samples. This experimental setting closely corresponds to the ideal scenario illustrated in our 2D simulation (Fig. 3(c)). When $\kappa$ approaches infinity, $x_0$ and $x_1$ become directionally identical, reducing the task to simple norm matching. Conversely, when $\kappa = 0$, the distribution becomes equivalent to Gaussian according to Section 3.1.

As shown in Table 3, while $\kappa$ approaching infinity yields FID scores near zero, this result is trivial since it merely reproduces the training data with norm scaling. More importantly, at reasonable $\kappa$ values (1500, 3000), we observe that multiple generations from the same $\text{vMF}(u_1, \kappa)$ distribution produce images that are structurally similar yet vary in fine details. This demonstrates that the model learns to generate diverse samples around each data point rather than simply memorizing individual training examples (see Fig. 9 in appendix). Even under oracle conditions, the model generates novel images not present in the original distribution, confirming the potential of directional alignment strategies.

However, this approach has a critical limitation: it requires storing the entire training dataset to generate corresponding direction-aligned source samples, making it impractical for large-scale applications.

### 5.2.2 Clustering-based Approach

The oracle pairing method, while effective, is impractical due to its massive storage requirements. To address this, we propose a more scalable alternative: clustering-based pairing. This method approximates the oracle setup by grouping similar data points and assigning them a shared source distribution, drastically reducing the computational and storage overhead.

The process is straightforward. First, we partition the normalized target data into $K$ groups using spherical $K$-means clustering (MacQueen, 1967). For each of the $K$ clusters, we then construct a vMF distribution as $\text{vMF}(c_j/\|c_j\|, \kappa)$, where $c_j$ is the centroid of cluster $j$, and sample source points following Eq. (7). Each target sample $x_1$ is then paired with a source sample $x_0$ drawn from the vMF corresponding to its assigned cluster. While this pairing is suboptimal compared to the oracle pairing, it offers a practical compromise between random Gaussian pairing and perfect directional alignment for suitable choices of $K$ and $\kappa$.

In Table 3, we benchmark the performance across a range of concentrations $\kappa$, fixing $K = 3$ found by an elbow study with silhouette scores. As the concentration parameter $\kappa$ decreases, each distribution covers a broader angular range. When $\kappa = 0$, the distribution becomes nearly identical to Gaussian, and generation performance is correspondingly similar. For moderately concentrated ranges of $\kappa \in \{50, 100\}$, we observe notable performance improvements. At high concentrations ($\kappa \geq 300$), however, performance drops below that of the Gaussian baseline. This degradation, similar to what we observed in Fig. 3(b), stems from path entanglement due to support deficiency as discussed in Section 4.2. When the source distribution becomes too concentrated, trajectories from nearby source points become entangled, making the vector field difficult to learn and causing training instability.

This finding highlights a fundamental trade-off: while directional alignment offers advantages, the source distribution must retain sufficient angular support to avoid trajectory interference. Without oracle-level pairing that guarantees consistent and perfect alignment, overly concentrated distributions create more problems than they solve, confirming that robust flow learning requires balanced coverage rather than extreme concentration.

## 6 Proposed Method

In this section, we introduce two complementary strategies based on previous findings. We first propose "Pruned Sampling", which creates a more efficient source distribution by identifying and removing directions that are directionally irrelevant to the target data. We then introduce "Norm Alignment" to resolve the scale mismatch between the source and target distributions. By creating a data-informed source, these methods improve the performance and stability of flow matching models without requiring architectural changes or retraining.

### 6.1 Pruned Sampling

Our previous experiments showed that we need to maintain sufficient support, but it is difficult to determine how much support is needed. Instead of attempting to find this amount, we propose the opposite strategy: pruning the Gaussian. The core idea is to start with a Gaussian source that covers all directions, then identify and prune away regions that are directionally irrelevant to the target data. By removing directions where data is absent, we can adjust the source distribution to better align with the target manifold.

While several approaches can be employed to identify directions where data is absent or sparse, we utilize Principal Component Analysis (PCA) (Pearson, 1901) for simplicity. Specifically, given a dataset $\mathcal{D} = \{x_i\}_{i=1}^N \subset \mathbb{R}^d$, we first $L_2$-normalize each sample by $\tilde{x}_i = x_i/\|x_i\|_2$ and compute an orthonormal basis $V = [v_1, \ldots, v_d] \in \mathbb{R}^{d \times d}$ via PCA, where the basis vectors are ordered by their corresponding eigenvalues in descending order. To account for directional symmetry, we construct an extended basis by including the opposite directions:

$$V_{\text{ext}} = [v_1, \ldots, v_d, v_{d+1}, \ldots, v_{2d}] \in \mathbb{R}^{d \times 2d}, \tag{8}$$

where $v_{k-d} = -v_k$ for $k = d+1, \ldots, 2d$. For every basis vector $v_k \in V_{\text{ext}}$, we measure the largest cosine similarity $\gamma_k = \max_i \langle v_k, \tilde{x}_i \rangle$ for $k = 1, \ldots, 2d$. Directions with a low $\gamma_k$ indicate that they are remote from

Table 4: **Pruned Sampling.** Comparison of FID scores for I-CFM and OT-CFM under different training and inference source configurations on CIFAR10. Euler integration is used for all evaluations. The best scores are indicated in **bold**.

| Method | Train Source | Inference Source | FID (NFE=5) | FID (NFE=100) |
|---|---|---|---|---|
| I-CFM | Gaussian | Gaussian | 34.74 | 4.36 |
| | *Pruned* | *Pruned* | 34.71 | 4.17 |
| | Gaussian | *Pruned* | **28.99** | **3.95** |
| OT-CFM | Gaussian | Gaussian | 18.55 | 4.40 |
| | *Pruned* | *Pruned* | 20.47 | 4.18 |
| | Gaussian | *Pruned* | **17.29** | **4.10** |

the data distribution. We denote a set of such basis vectors $\mathcal{R} = \{k \in \{1, \ldots, 2d\} : \gamma_k \leq \tau\}$, where $\tau$ is a hyperparameter indicating the threshold. We then apply rejection sampling: we sample $x_0 \sim \mathcal{N}(0, I)$ and retain it if and only if $\max_{k \in \mathcal{R}} \left\langle v_k, \frac{x_0}{\|x_0\|_2} \right\rangle \leq \tau_r$, where $\tau_r > \tau$ provides a more conservative margin to prevent potential support deficiency in critical directions. Detailed experimental settings can be found in Sections F and G.

Based on this approach, we compare three strategic approaches: (i) Gaussian→Gaussian, where the model is trained and sampled from the complete Gaussian distribution; (ii) Pruned→Pruned, which involves training and sampling exclusively from directions that meet the pruning criteria; and (iii) Gaussian→Pruned where the model is trained on the complete Gaussian distribution but sampled only from its pruned subset.

Empirical results in Table 4 reveal unexpected patterns. The fully pruned method (ii) yields inconsistent performance compared to the Gaussian baseline (i). While it offers marginal improvements at a high number of function evaluations (NFE), it has minimal impact on I-CFM at low NFE and negatively affects the performance of OT-CFM. In contrast, the hybrid method (iii) consistently outperforms the other methods across all evaluated configurations.

Although this observation may seem counterintuitive, it aligns with our 2D simulation findings in Section 4.3. The effectiveness of the hybrid "Gaussian→Pruned" strategy stems from the distinct requirements of the training and inference phases.

During training, leveraging the full Gaussian source is advantageous. A high-dimensional Gaussian distribution provides uniform angular coverage, ensuring the model learns the vector field omnidirectionally around data modes. This comprehensive training is critical for building a robust model that can generalize across diverse initialization conditions. Conversely, restricting training to only the pruned directions ($\mathcal{S}$) limits the model's exposure. Without supervision from the excluded directions ($\mathbb{S}^{d-1} \setminus \mathcal{S}$), the model's ability to generalize to unseen trajectories is compromised. This limitation is particularly detrimental at low NFEs, where the model has fewer opportunities to correct for poor initializations.

During inference, however, certain regions of the Gaussian source are suboptimal for sampling. These include areas devoid of data and singular points equidistant from multiple data modes. Source samples ($x_0$) from these regions are surrounded by fewer interpolation paths, leading to less frequent updates during training (see Fig. 5). Consequently, the vector field near these samples is learned less accurately, making them prone to generating erroneous outputs. Pruning these directions from source distribution during inference effectively guides the sampling process toward more reliable paths, enhancing generation quality.

In summary, the optimal strategy involves training on the complete Gaussian distribution to ensure the model learns a comprehensive and generalizable vector field, while sampling exclusively from the pruned subset to avoid regions where the learned field is likely to be inaccurate.

This finding has important practical implications. Pruned Sampling can be applied as a post-processing to *any* pretrained flow matching model that uses a Gaussian source distribution, requiring no retraining or architectural modifications. We demonstrate the effectiveness and scalability of our approach across multiple settings. In Table 5, applying Pruned Sampling to a higher-dimensional ImageNet64 pretrained model yields consistent performance improvements, confirming the scalability of our approach. Furthermore, Table 6

Table 5: FID scores on ImageNet64 using Euler integration for OT-CFM w/wo source *Pruned*.

| Model | Source | NFE | | | |
|---|---|---|---|---|---|
| | | **5** | **10** | **20** | **100** |
| OT-CFM | Gaussian | 53.95 | 18.84 | 10.62 | 9.10 |
| | *Pruned* | **49.56** | **16.70** | **9.54** | **8.78** |

Table 6: FID scores on CIFAR10 using Euler integration for various numbers of function evaluations (NFE) for OT-CFM and I-CFM, comparing the baseline (Gaussian source) with our proposed method that incorporates *Pruned* and/or *NormAlign*. Our methods are indicated in *italics*. Reported values are mean $\pm$ standard deviation.

| Train Method | Source | NFE | | | | |
|---|---|---|---|---|---|---|
| | | **5** | **10** | **20** | **40** | **100** |
| OT-CFM (Baseline) | Gaussian | $19.80 \pm 0.39$ | $10.95 \pm 0.34$ | $7.41 \pm 0.20$ | $5.61 \pm 0.10$ | $4.40 \pm 0.01$ |
| OT-CFM | *Pruned* | $\mathbf{17.24 \pm 0.08}$ | $\mathbf{9.17 \pm 0.14}$ | $6.34 \pm 0.05$ | $4.92 \pm 0.11$ | $4.15 \pm 0.07$ |
| OT-CFM + *NormAlign* | Gaussian | $24.92 \pm 0.17$ | $11.39 \pm 0.01$ | $6.81 \pm 0.07$ | $4.98 \pm 0.06$ | $4.03 \pm 0.08$ |
| OT-CFM + *NormAlign* | *Pruned* | $21.52 \pm 0.22$ | $9.55 \pm 0.06$ | $\mathbf{5.85 \pm 0.03}$ | $\mathbf{4.53 \pm 0.06}$ | $\mathbf{3.88 \pm 0.03}$ |
| I-CFM (Baseline) | Gaussian | $34.49 \pm 0.24$ | $13.30 \pm 0.06$ | $7.75 \pm 0.08$ | $5.63 \pm 0.06$ | $4.35 \pm 0.02$ |
| I-CFM | *Pruned* | $\mathbf{28.78 \pm 0.19}$ | $\mathbf{10.37 \pm 0.15}$ | $\mathbf{6.40 \pm 0.06}$ | $4.89 \pm 0.09$ | $3.97 \pm 0.03$ |
| I-CFM + *NormAlign* | Gaussian | $52.20 \pm 0.09$ | $18.94 \pm 0.18$ | $8.10 \pm 0.04$ | $4.93 \pm 0.08$ | $3.79 \pm 0.02$ |
| I-CFM + *NormAlign* | *Pruned* | $48.10 \pm 0.34$ | $16.58 \pm 0.24$ | $7.04 \pm 0.06$ | $\mathbf{4.56 \pm 0.03}$ | $\mathbf{3.64 \pm 0.02}$ |

shows that Pruned Sampling consistently enhances performance for both OT-CFM and I-CFM, regardless of the number of function evaluations (NFE). By simply filtering out initialization points from data-sparse regions during inference, existing models can achieve immediate performance improvements with minimal computational overhead.

## 6.2 Norm Alignment Strategy

In addition to the directional considerations, we identify another fundamental disparity in flow matching: the significant difference in scale between the source and target distributions. Flow matching models conventionally operate between a standard Gaussian source $\mathcal{N}(0, I)$ and the target data distributions, creating a substantial norm disparity that impacts model performance. The standard Gaussian source samples typically exhibit norms of approximately $\sqrt{d}$, while target data samples (often normalized to $[-1, 1]^d$) possess norms considerably smaller. This disparity necessitates significant computational resources—either in model capacity or training iterations—to resolve, potentially diverting attention from more crucial aspects of distribution estimation.

To address this disparity, we propose a *Norm Alignment* strategy. Specifically, we compute the expected norm of the target distribution, $E_{x_1 \sim p_1}[\|x_1\|]$, and that of the source distribution, $E_{x_0 \sim p_0}[\|x_0\|]$, which is $E[\chi(d)]$ when $p_0 = \mathcal{N}(0, I)$. We scale the target samples to $x_1' \equiv x_1 \cdot E[\|x_0\|]/E[\|x_1\|]$, placing the transformed target distribution $p_1'$ on a hypersphere with a radius matching the average norm of $p_0$. During inference, we reverse this scaling by multiplying $x_1$ by $E[\|x_1\|]/E[\|x_0\|]$, thus recovering the original scale of the target samples.

Aligning the average norms of source and target samples through the proportional scaling described above yields substantial performance gains, as demonstrated in Table 6. This suggests that flow models face greater difficulty than expected in learning to resolve norm mismatches, requiring significant computational resources that could otherwise be devoted to more essential aspects of the generation task. However, at low numbers of function evaluations (NFE), applying *Norm Alignment* can actually degrade performance. This is because, when both source and target distributions lie on the same norm hypersphere, some transport trajectories are restricted to move close to the surface of this hypersphere. As a result, the transport paths become more curved, requiring a greater number of NFEs for the model to accurately follow these paths and achieve effective transport. In contrast, without norm alignment, source samples can move more directly toward

lower-norm regions near the data manifold, resulting in straighter transport paths at low NFEs. A more detailed discussion and visualization of this phenomenon can be found in Section E.

As a final remark, combining *Norm Alignment* and *Pruned Sampling* at NFE 100 on CIFAR10 dataset yields substantial improvements: reducing FID by 0.67 for OT-CFM and 0.72 for I-CFM. These results highlight the importance of addressing both directionality and norm alignment in the source distribution to fully unlock the potential of flow matching models.

## 7 Conclusion

This work investigates whether alternative source distributions can outperform the standard Gaussian in flow matching. To better understand how flow matching models learn, we propose a novel 2D simulation designed to visualize and interpret high-dimensional behaviors in flow matching. Our simulations show strong agreement with actual high-dimensional results, highlighting the effectiveness of this approach for analyzing and gaining insights into flow matching.

Our analysis reveals that intuitive strategies often fail. Density approximation approaches, using models like GMMs or CNFs, suffer from mode discrepancy: by omitting low-density target modes, they force the model to learn inefficient pathways. Similarly, directional alignment methods induce path entanglement when the source becomes too concentrated, destabilizing optimization. We find that the success of the Gaussian distribution lies in its omnidirectional coverage, which ensures robust vector field learning around all data modes.

Building on these insights, we introduce a hybrid framework that combines Norm Alignment during training with Pruned Sampling at inference. This preserves the robustness of Gaussian-based training while eliminating problematic initializations from data-sparse regions. Crucially, Pruned Sampling can be applied to pre-trained models without any retraining, offering a significant practical advantage. Our extensive experiments confirm consistent improvements across multiple settings, demonstrating both the practical value of our insights and the effectiveness of our readily applicable technique for advancing flow matching.

## Acknowledgment

This work was supported by Youlchon Foundation, NRF grants (RS-2021-NR05515, RS-2024-00336576, RS-2023-0022663) and IITP grants (RS-2022-II220264, RS-2024-00353131) by the government of Korea.

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

## Appendix

## A    Derivation of the $\chi$-Sphere Decomposition

Let $x \sim \mathcal{N}(0, I)$ with density

$$f(x) = \frac{1}{(2\pi)^{d/2}} \exp\left(-\tfrac{1}{2}\|x\|_2^2\right).$$

**Change of variables.**    Let us denote $x = rs$, where $r = \|x\|_2 \in [0, \infty)$ and $s = x/\|x\|_2 \in S^{d-1}$. The mapping $(r, s) \mapsto x$ is one-to-one on $(0, \infty) \times S^{d-1}$ with Jacobian determinant $|\det J| = r^{d-1}$, giving the joint density

$$f_{r,s}(r, s) = \frac{1}{(2\pi)^{d/2}} e^{-r^2/2} \, r^{d-1}, \qquad r > 0, \ s \in S^{d-1}.$$

**Marginal of $s$.**    Integrating $f_{r,s}$ over $r$ yields

$$f_s(s) = \int_0^\infty f_{r,s}(r, s) \, dr = \frac{1}{(2\pi)^{d/2}} \int_0^\infty e^{-r^2/2} r^{d-1} \, dr = \frac{1}{\omega_{d-1}},$$

because the integral equals to $(2\pi)^{d/2}/\omega_{d-1}$, where $\omega_{d-1} = 2\pi^{d/2}/\Gamma(\tfrac{d}{2})$ is the surface area of $S^{d-1}$. Hence, $s \sim \mathcal{U}(S^{d-1})$, where $\mathcal{U}(S^{d-1})$ denotes the uniform distribution on the unit sphere.

**Marginal of $r$.**    Integrating $f_{r,s}$ over the sphere and using $\int_{S^{d-1}} ds = \omega_{d-1}$ gives

$$f_r(r) = \omega_{d-1} \, f_{r,s}(r, s) = \frac{1}{2^{d/2-1}\Gamma(\tfrac{d}{2})} \, r^{d-1} e^{-r^2/2},$$

which is the probability density function of the $\chi_d$ distribution.

Because $f_{r,s}$ factorizes as $f_r(r)f_s(s)$, $r$ and $s$ are independent. Therefore, $x \stackrel{d}{=} rs_G$ with $r \sim \chi(d)$ and $s_G \sim \mathcal{U}(S^{d-1})$.

## B    Implementation Details and Results for 2D Simulations

Table 7: **Performance Comparison in 2D Simulation.** Quantitative evaluation of flow matching performance across different source distribution strategies and pairing methods in 2D simulations. Results are averaged over 10 independent runs with standard deviations. Lower values indicate better performance for all metrics.

| Source | Method | AMD ↓ | Failure Rate ↓ | MDD ↓ | Norm. Wasserstein ↓ |
|---|---|---|---|---|---|
| Gaussian ($\chi$-Sphere) | I-CFM | $0.19 \pm 0.03$ | $1.88 \pm 0.73$ | $0.100 \pm 0.031$ | $1.45 \pm 0.15$ |
| Gaussian ($\chi$-Sphere) | OT-CFM | $0.29 \pm 0.05$ | $4.35 \pm 1.60$ | $0.066 \pm 0.028$ | $1.50 \pm 0.19$ |
| Pruned Gaussian ($\chi$-Sphere) | I-CFM | $0.18 \pm 0.04$ | $1.40 \pm 0.91$ | $0.128 \pm 0.056$ | $1.36 \pm 0.12$ |
| Flow Model 200-iter | OT-CFM | $0.31 \pm 0.09$ | $4.99 \pm 2.32$ | $0.056 \pm 0.014$ | $1.34 \pm 0.22$ |
| Flow Model 6k-iter | OT-CFM | $0.32 \pm 0.05$ | $6.05 \pm 1.24$ | $0.017 \pm 0.005$ | $1.40 \pm 0.18$ |
| Flow Model 10k-iter | OT-CFM | $0.32 \pm 0.02$ | $6.14 \pm 1.30$ | $0.020 \pm 0.009$ | $1.47 \pm 0.17$ |
| Directional Align (Normal) | Perfect OT | $0.16 \pm 0.01$ | $0.93 \pm 0.15$ | $0.001 \pm 0.001$ | $0.52 \pm 0.05$ |
| Directional Align (Normal) | OT-CFM | $0.18 \pm 0.02$ | $1.80 \pm 0.51$ | $0.016 \pm 0.008$ | $1.03 \pm 0.22$ |
| Directional Align (Tight) | OT-CFM | $0.26 \pm 0.02$ | $4.08 \pm 0.68$ | $0.012 \pm 0.004$ | $1.15 \pm 0.17$ |

**Implementation Details.**    Our 2D simulations are implemented using PyTorch with the TorchCFM library (Tong et al., 2024). The vector field model employs a multi-layer perceptron (MLP) architecture with time-varying inputs, trained using the Adam optimizer for 20,000 iterations with a batch size of 16. The network consists of fully connected layers with hidden dimensions carefully chosen to capture the flow dynamics in 2D space.

To simulate high-dimensional geometric properties within our 2D framework, we implement the $\chi$-Sphere decomposition as follows. For the source distribution, we sample radii from a chi distribution with $d = 3072$ degrees of freedom to approximate high-dimensional norm concentration, then uniformly sample directions $\theta \in [0, 2\pi]$. For the target distribution, we construct three directional clusters with varying densities to reflect realistic data scenarios. The clusters are positioned at base angles $[4.4956, 5.9167, 1.1018]$ radians with corresponding density ratios $[0.05, 0.3, 0.65]$. Each cluster is generated by sampling angles around the base direction with small perturbations ($\pm 10°$ plus additional Gaussian noise) and radii from $\chi(625)$.

Trajectory visualization is performed every 5,000 training iterations using the Dormand-Prince ODE solver with adaptive step size control (absolute tolerance $= 1 \times 10^{-4}$, relative tolerance $= 1 \times 10^{-4}$). We classify trajectories as successful or failed based on the L2 distance between generated samples and the nearest target data point, using a threshold of 1.0 unit. Successful trajectories (distance $\leq 1.0$) are visualized in light steel blue, while failed trajectories (distance $> 1.0$) appear in indian red, providing clear visual feedback on the effectiveness of different source distribution strategies.

**Evaluation Metrics.** Average Minimum Distance (AMD) measures generation accuracy by computing the mean L2 distance between each generated sample and its nearest target data point. This metric directly quantifies how closely the generated samples align with the true data distribution, with lower values indicating better approximation quality. Failure Rate quantifies the proportion of generated samples that exceed a distance threshold of 1.0 unit from any target data point. This binary metric captures the frequency of catastrophic generation failures where samples fall completely outside the data manifold, providing insight into the robustness of different source distribution strategies. Mode Distribution Divergence (MDD) assesses structural fidelity by measuring the KL-divergence between the mode distributions of generated and true data samples. We compute this by first assigning each sample to its nearest data mode cluster, then calculating the divergence between the resulting mode proportion vectors. This metric is particularly sensitive to mode collapse or imbalanced mode coverage, making it crucial for evaluating multimodal generation quality. Normalized Wasserstein quantifies the overall discrepancy between generated and true data distributions using the Wasserstein-1 distance, normalized by the average pairwise distance within the true data distribution. This normalization makes the metric robust to scale differences and enables meaningful comparison across different experimental configurations. Unlike standard Wasserstein distance, this normalized variant is specifically designed to handle cases where data consists of multiple modes with potentially imbalanced proportions, providing a more reliable measure of distributional similarity in complex, multimodal settings.

## C   Derivation of Path-Entanglement Bound

**Setup.**   Let $s_0, s_0' \overset{\text{i.i.d.}}{\sim} \text{Cap}(\mu, \gamma)$ be unit-norm directions drawn from the *spherical-cap* distribution

$$\text{Cap}(\mu, \gamma) := \left\{ s \in \mathbb{S}^{d-1} \ \middle|\ s^\top \mu \geq \gamma \right\}, \qquad 0 < \gamma < 1.$$

Writing $\theta = \arccos(s_0^\top s_0')$, we have

$$\mathbb{E}[1 - s_0^\top s_0'] = \mathcal{O}(1 - \gamma), \qquad \mathbb{E}[\theta] = \mathcal{O}((1 - \gamma)^{1/2}). \tag{A.1}$$

**Bounding the initial Euclidean separation.**   Draw radii $r, r' \overset{\text{i.i.d.}}{\sim} \chi(d)$ and set $x_0 = r s_0$, $x_0' = r' s_0'$. Because $\mathbb{E}[r] \simeq \sqrt{d}$, we have

$$\|x_0 - x_0'\|^2 = r^2 + r'^2 - 2rr' s_0^\top s_0' \approx 2d(1 - s_0^\top s_0'),$$

so that

$$\mathbb{E}\|x_0 - x_0'\| = \Theta((1 - \gamma)^{1/2}). \tag{A.2}$$

**Along the linear interpolation.**   For the straight-line paths $x_t = (1 - t)x_0 + tx_1$, $x_t' = (1 - t)x_0' + tx_1'$, let $d(t) = x_t - x_t'$. Then

$$\min_{t \in [0,1]} \|d(t)\| \leq \|x_0 - x_0'\| = \mathcal{O}((1 - \gamma)^{1/2}). \tag{A.3}$$

**Implication for the local Lipschitz constant.** A finite-difference estimate of the learned vector-field's Lipschitz constant yields

$$L_{\text{local}} \gtrsim \frac{\left\| v_\theta(x_t, t) - v_\theta(x'_t, t) \right\|}{\|x_t - x'_t\|} \approx \frac{\sin\theta}{\|x_1 - x'_1\| + \mathcal{O}\big((1-\gamma)^{1/2}\big)}, \qquad \theta = \cos^{-1}\!\Big(\frac{(x_1 - x_0)^\top (x'_1 - x'_0)}{\|x_1 - x_0\| \, \|x'_1 - x'_0\|}\Big). \quad (A.4)$$

As $\gamma \to 1$, the $(1-\gamma)^{1/2}$ term dominates the denominator, so $L_{\text{local}} \to \infty$, capturing the *path-entanglement* effect.

## D  DCT Refinement

We designed the DCT masks to filter out unnecessary high frequency details from both the source distribution and the target distribution.

### D.1  DCT Block

The Discrete Cosine Transform (DCT) (Ahmed et al., 1974; Strang, 1999) decomposes an image into a set of cosine basis functions with varying two-dimensional spatial frequencies. In practice, the 2D DCT is commonly applied to zero-centered image blocks $b$ of size $D \times D$, producing a corresponding $D \times D$ DCT coefficient matrix B.

$$B_{u,v} = \alpha_u \alpha_v \sum_{x=0}^{D-1} \sum_{y=0}^{D-1} b_{x,y} \cos\left(\frac{(2x+1)u\pi}{16}\right) \cos\left(\frac{(2y+1)v\pi}{16}\right),$$

$$\alpha_u = \begin{cases} \frac{1}{\sqrt{D}} & \text{if } u = 0, \\ \sqrt{\frac{2}{D}} & \text{if } u \neq 0, \end{cases} \qquad \alpha_v = \begin{cases} \frac{1}{\sqrt{D}} & \text{if } v = 0, \\ \sqrt{\frac{2}{D}} & \text{if } v \neq 0. \end{cases} \quad (9)$$

where $x$ and $y$ denote the horizontal and vertical pixel coordinates, while $u$ and $v$ are indices of corresponding horizontal and vertical spatial frequencies. $\alpha$ serves as a normalization term to ensure the orthogonality of the DCT basis functions. Following JPEG (Wallace, 1992) compression standard, $D = 8$ splitting the whole image into $8 \times 8$ blocks. Then, each blocks is converted to the $YCbCR$ color space, which separates image content into one luminance channel $Y$, representing brightness, and two chrominance channels $Cb$ and $Cr$, representing color information. Subsequently, the DCT is applied independently to each $D \times D$ block within all three channels.

### D.2  Data dependent DCT mask

To construct dataset-specific DCT-based frequency masks, we first applied the Discrete Cosine Transform (DCT) to every $D \times D$ block in the CIFAR-10 dataset. We computed the mean and standard deviation of each DCT coefficient across all blocks and across the three YCbCr channels: luminance (Y), blue chrominance (Cb), and red chrominance (Cr). Based on these statistics, we defined two types of frequency masks: a weak mask, which retains coefficients with absolute mean less than 0.1 and absolute standard deviation less than $\sqrt{2}$; and a strong mask, which allows a broader range of coefficients with standard deviation less than 2 under the same mean threshold. Notably, the luminance channel was excluded from the masking process. This decision was based on two factors: (1) its coefficients did not meet the criteria required for masking, and (2) luminance plays a critical role in human visual perception, making its full preservation essential for maintaining perceptual quality. These masks were applied uniformly to both the Gaussian samples and the ground-truth targets to improve alignment and suppress irrelevant high-frequency components.

The weak DCT-based mask achieved a slightly improved FID score compared to the baseline using a naïve Gaussian source distribution, as shown in Table 8. This improvement is attributed to the suppression of unnecessary high-frequency details in both the source and the target, which promotes better alignment. However, the strong DCT mask led to a decline in performance relative to the Gaussian baseline, suggesting

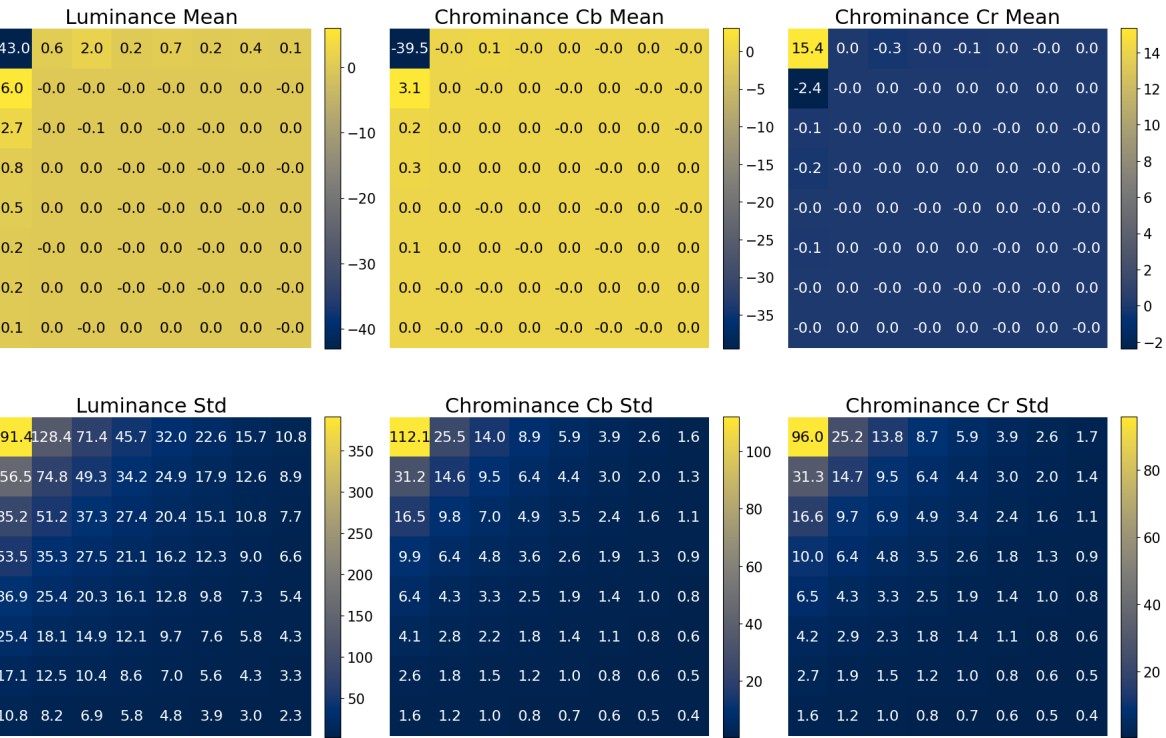

Figure 6: Mean and standard deviation of DCT coefficients computed across all $D \times D$ blocks in the CIFAR-10 dataset, separately for the Y (luminance), Cb (blue-difference chrominance), and Cr (red-difference chrominance) channels. These statistics were used to construct the frequency masks by identifying coefficients with low absolute mean and variance.

Table 8: FID scores for different masking strategies. The DCT-Weak and DCT-Strong methods apply frequency masking based on dataset-specific DCT coefficient statistics, while the Gaussian baseline applies no masking.

| Method | Gaussian | DCT-Weak | DCT-Strong |
|---|---|---|---|
| FID↓ | 4.30 | **4.20** | 4.39 |

limitations of this approach. Since the masking is applied symmetrically to both the source and the target, increasing the masking strength can degrade the fidelity of the target representation, ultimately reducing model performance.

# E   Explanation on why Norm Alignment Hurts at Low NFE

This issue stems from the fundamental change in transport geometry when both source and target distributions are constrained to similar norm hyperspheres. Under a standard flow matching without norm alignment, trajectories typically follow relatively straight paths from the outer Gaussian source toward the inner data manifold. However, when norm alignment is applied, both distributions exist on similar radii, fundamentally altering the transport dynamics.

As the transport operates within a batch with a finite number of samples, samples often tend to pair with targets from a region with higher density, even with optimal transport pairing. This causes the learned vector field to curve slightly toward other data-concentrated regions rather than following perfectly straight paths to the nearest points. When trajectories span from large to small radii (standard case), this pairing bias has minimal impact on path straightness. However, when both source and target exist on similar hyperspheres

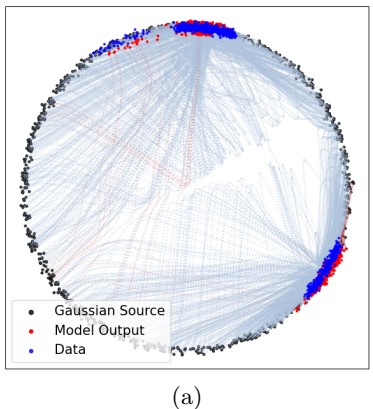 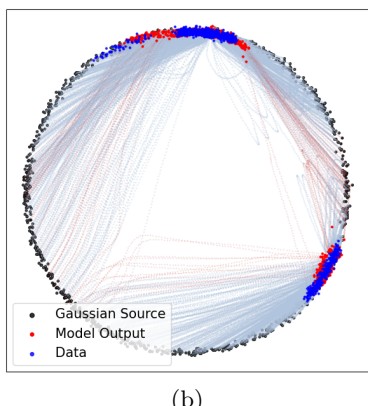

(a)                                                      (b)

Figure 7: **Flow matching visualizations with *Norm Alignment*.** (a) I-CFM with Gaussian source (b) OT-CFM with Gaussian source

(when norms are aligned), the same pairing bias results in curved trajectories as the vector field gets "pulled" laterally toward high-density regions rather than following direct radial paths.

This curvature becomes problematic at very low NFE, because ODE solvers require more integration steps to accurately approximate curved paths compared to straight ones. With insufficient steps, the solver's linear approximations between steps fail to capture the trajectory curvature, leading to accumulation of integration errors and degraded sample quality. As visualized in Fig. 7, the norm alignment fundamentally changes the transport geometry, causing trajectories to curve. This curvature is poorly approximated by ODE solvers with few steps, leading to the observed degradation in sample quality.

## F  Experimental Setup of High-dimensional Validation

We conduct experiments on CIFAR-10 (Krizhevsky & Hinton, 2009) and ImageNet64 (Deng et al., 2009). We adopt an improved UNet (Ronneberger et al., 2015) architecture with attention, following the ADM framework (Dhariwal & Nichol, 2021). Both models use a base channel width of 128, four attention heads with 64 channels each, Group Normalization, GELU activations, dropout rate of 0.1, and convolutional resampling. For CIFAR-10 ($32 \times 32$), the model uses 2 residual blocks per resolution and applies attention at $16 \times 16$. For ImageNet64 ($64 \times 64$), we use 3 residual blocks with attention at $32 \times 32$ and $16 \times 16$ resolutions.

Training is performed using the Adam optimizer with a learning rate of $2 \times 10^{-4}$ and a linear warmup schedule. For CIFAR-10, we train for 200K steps with a batch size of 512 and 5K warmup steps. For ImageNet64, we use 350K steps, a batch size of 384, and 20K warmup steps. All experiments are run on a single NVIDIA RTX A6000 (48GB) GPU.

## G  Source Pruning Settings

For source pruning, we empirically set the pruning thresholds to $\tau = 0.01$, $\tau_r = 0.048$ for CIFAR-10 and $\tau = 0.005$, $\tau_r = 0.026$ for ImageNet64.

To provide further insight into the effect of these hyperparameters, we conducted an ablation study on CIFAR-10 by varying the threshold $\tau_r$. Table 9 reports the Fréchet Inception Distance (FID) scores, total generation time for 512 images, and the corresponding rejection rates. The results illustrate a trade-off: as $\tau_r$ increases, total computation time decreases, but this is accompanied by a slight degradation in sample quality (higher FID). Nevertheless, our pruned sampling approach consistently outperforms the standard Gaussian sampling baseline ($\tau_r =1.0$) in FID, even at very low rejection rates with no additional computation.

Table 9: Ablation study on the pruning threshold $\tau_r$ for CIFAR-10. FID scores are reported for both ICFM and our OTCFM. The total time corresponds to generating 512 images.

| $\tau_r$ | FID (I-CFM / OT-CFM) | Total Time (s) | Single Image (ms) | Rejection Rate (%) |
|---|---|---|---|---|
| 0.05 | 4.00 / 4.10 | 19.96 | 38.98 | 99.94 |
| 0.055 | 4.05 / 4.10 | 18.11 | 35.37 | 94.90 |
| 0.06 | 4.13 / 4.21 | 18.09 | 35.32 | 70.08 |
| 0.07 | 4.22 / 4.28 | 18.09 | 35.32 | 11.88 |
| 0.08 | 4.24 / 4.33 | 18.09 | 35.32 | 0.97 |
| 1.0 (Gaussian) | 4.36 / 4.40 | 18.09 | 35.32 | 0.00 |

## H    Norm Scaling of 2D Simulations

As described in Section 3.2, we scale the approximated source so that its average norm matches the average radius of the $\chi$-Sphere. This normalization ensures that all source types start from a comparable average norm, allowing us to focus on visualizing the transport path patterns relative to the $\chi$-Sphere. While this norm scaling may reduce the apparent convergence of the source to the data, it provides a clearer and more consistent visualization of the transport dynamics across different source settings. Without this normalization, the dynamics become difficult to interpret, as demonstrated in Fig. 8.

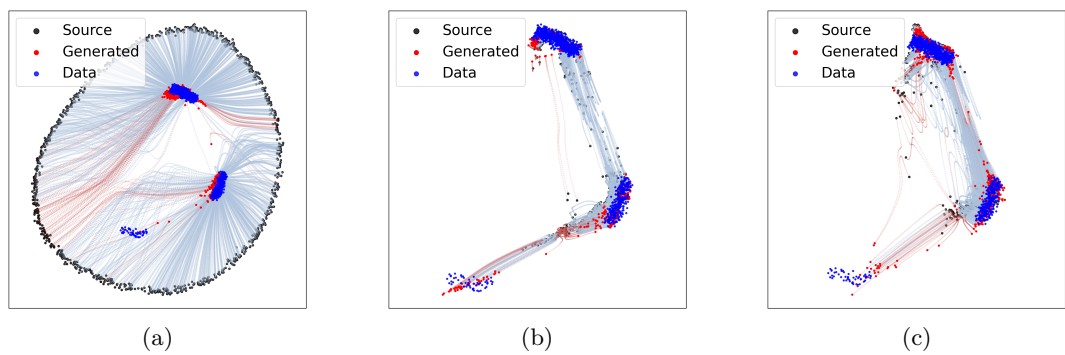

(a)                                    (b)                                    (c)

Figure 8: **Visualization of flow matching with density-approximated source without norm scaling.** (a) OT-CFM with Approximated Source (iter 200), (b) OT-CFM with Approximated Source (iter 6000), (c) OT-CFM with Approximated Source (iter 10000).

## I    Limitations

While our study provides a conceptual understanding of the learning dynamics of flow matching and valuable insights into the design of source distributions for flow matching, several limitations remain.

First, our conclusions primarily derive from experiments on image datasets such as CIFAR-10 and ImageNet64. These datasets may not fully capture the challenges of other modalities like text, audio, or molecular data, where distribution geometries and semantics differ significantly. This also applies to latent spaces, such as those from image tokenizers (Rombach et al., 2022; Esser et al., 2024; Podell et al., 2023; Yao et al., 2025; Chen et al., 2025; Lee et al., 2025; Chen et al., 2024), where the structure of embeddings can vary widely. Recent studies investigating the geometric properties of latent spaces—such as analyzing intrinsic manifold curvature (Arvanitidis et al., 2017; Choi et al., 2021) or learning structured, isometric representations (Preechakul et al., 2022; Hahm et al., 2024)—highlight the complexity of these domains. The generalizability of our findings to these domains remains an open question. Second, our analysis of several source distributions offers practical insights but falls short of establishing a comprehensive theoretical

foundation for optimal source distribution design in high-dimensional spaces. Further work is needed to develop rigorous guarantees and theoretical bounds that can guide more principled approaches to source distribution optimization across diverse application domains. Third, while source pruning and norm alignment improve performance, they introduce additional hyperparameters—such as pruning thresholds and scaling factors—that require careful tuning and incur additional computational overhead due to rejection sampling. Moreover, their effectiveness is sensitive to the number of function evaluations (NFEs). Notably, norm alignment may degrade performance in very low-NFE regimes, presumably due to increased curvature in the generative trajectory. Finally, our study focuses on unconditional generation. The applicability of our findings to conditional generation tasks, such as class-conditional or text-to-image generation, remains unexplored. Various conditioning mechanisms can interact with the source distribution design to change the effectiveness of the proposed strategy, which requires future research.

Despite these limitations, our work establishes important empirical guidelines for source distribution design and provides a foundation for future research in this area. We believe that addressing these limitations through expanded empirical evaluation, theoretical analysis, and computational optimization will be valuable directions for advancing flow matching methods.

## J   Qualitative examples

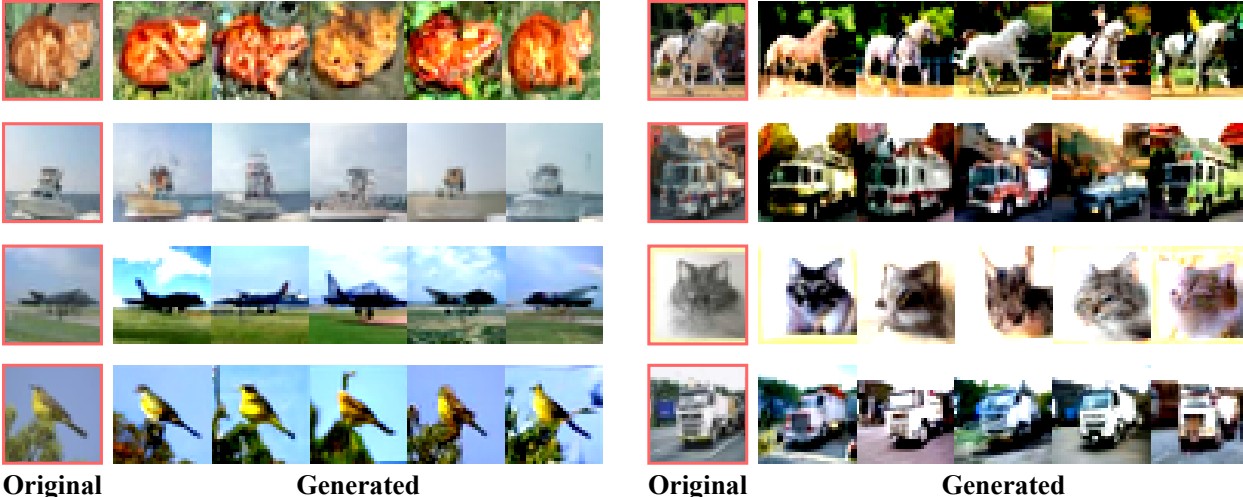

**Original**          **Generated**          **Original**          **Generated**

Figure 9: Left-most images (red frame) of each cluster are target samples $x_1$; the five images on the right are generated from $x_0 \sim \text{vMF}(\mu(x_1), \kappa = 1500)$ by the generator. Although direct pairing to concentrated areas raised concerns about potential overfitting to the target data, we observe that the generated images remain diverse and visually distinct while being similar.

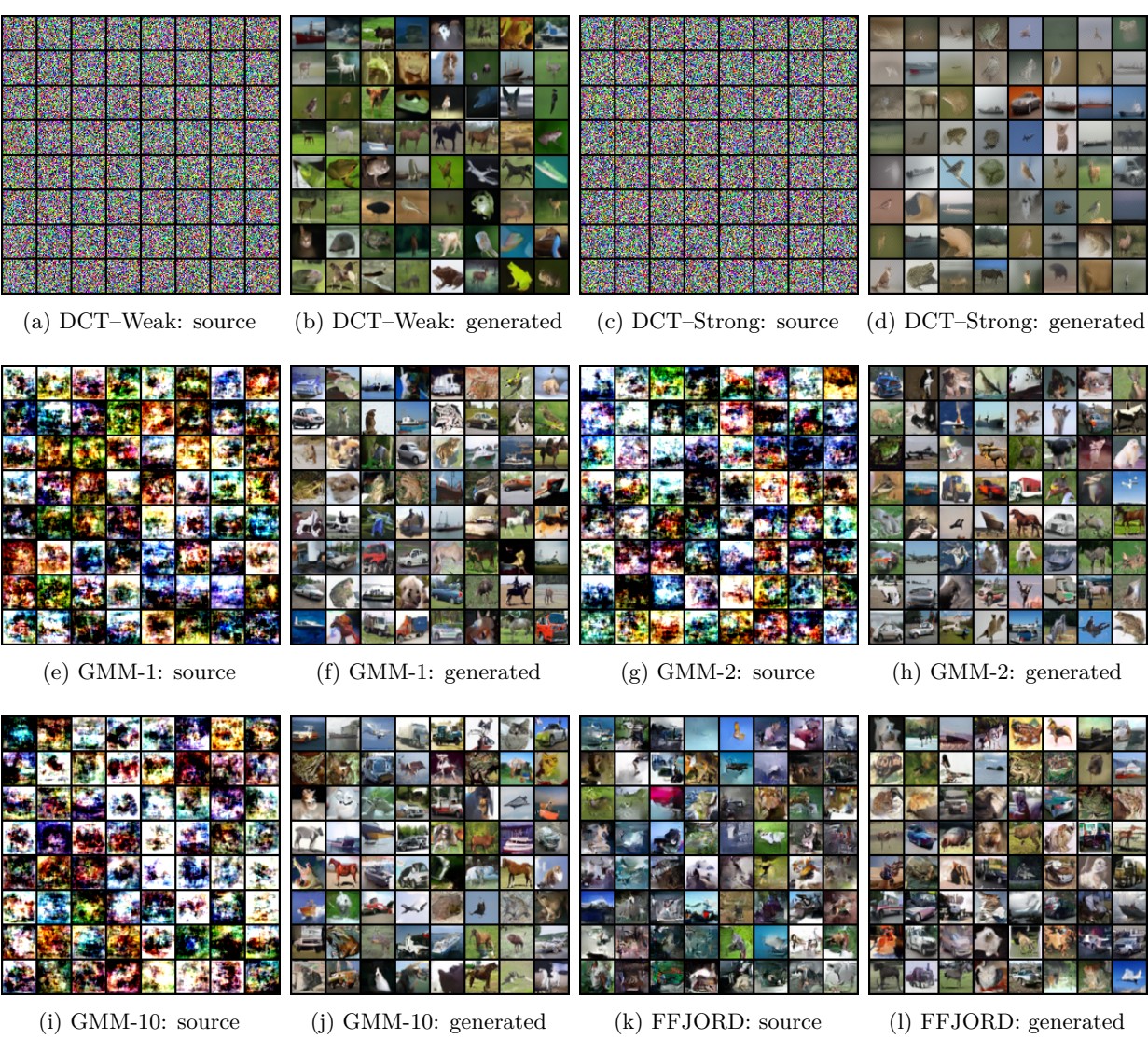

(a) DCT–Weak: source     (b) DCT–Weak: generated     (c) DCT–Strong: source     (d) DCT–Strong: generated

(e) GMM-1: source     (f) GMM-1: generated     (g) GMM-2: source     (h) GMM-2: generated

(i) GMM-10: source     (j) GMM-10: generated     (k) FFJORD: source     (l) FFJORD: generated

Figure 10: **Source–vs.–generated 2-D samples for discrete mixtures (Part I).** Each adjacent pair visualises the original training distribution (*left*) and the output of the generator (*right*) for DCT-Weak, DCT-Strong, three GMM settings, and FFJORD.

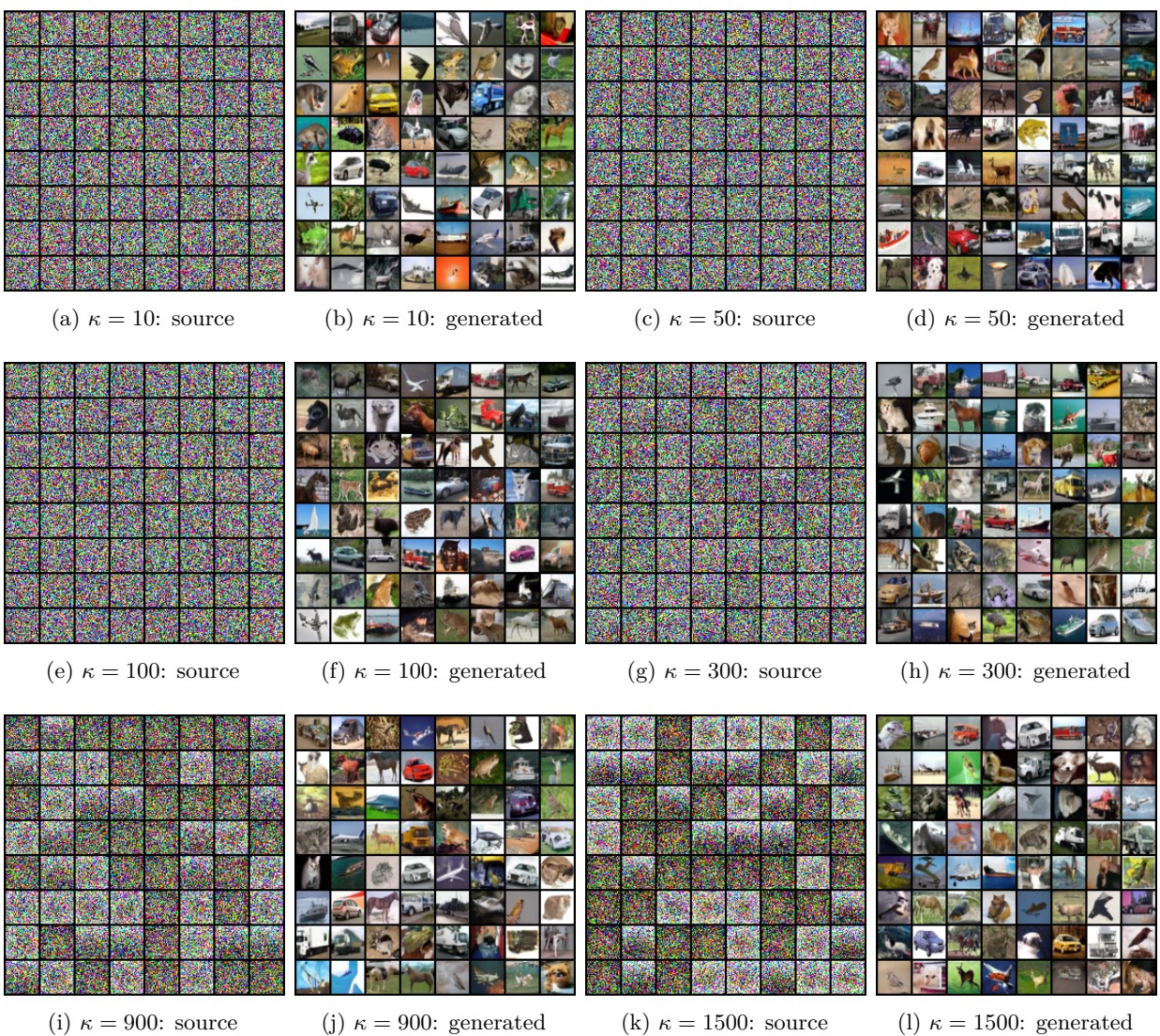

(a) $\kappa = 10$: source  (b) $\kappa = 10$: generated  (c) $\kappa = 50$: source  (d) $\kappa = 50$: generated

(e) $\kappa = 100$: source  (f) $\kappa = 100$: generated  (g) $\kappa = 300$: source  (h) $\kappa = 300$: generated

(i) $\kappa = 900$: source  (j) $\kappa = 900$: generated  (k) $\kappa = 1500$: source  (l) $\kappa = 1500$: generated

Figure 11: **Source–vs.–generated 2-D samples for von Mises–Fisher distributions (Part II).** Pairs compare the empirical data (*left*) with model outputs (*right*) at increasing concentration parameters $\kappa$ from 10 to 1500.

