# OpenReview forum: "Is There a Better Source Distribution than Gaussian? Exploring Source Distributions for Image Flow Matching"
_TMLR — Accepted by TMLR_

### Review · Reviewer_7yfb · 2025-08-18

**Summary Of Contributions:**

The paper contributes to the expanding literature of shaping (non-gaussian) source distributions for flow-matching in a data-driven manner.

Specifically, the authors explore a number of cases, both synthetic and realistic which reflect complex high-dimensional geometries.

The synthetic dataset is constructed is two-dimensional, but designed to mimic high-dimensional geometry through a spherical coordinate decomposition - with 3 clusters of data of unequal size used to mimic complex data in typical high-dimensional settings.

Based on various experiments using these datasets, the authors explore various strategies on exploiting data-geometry to identify better source distributions.   They conclude the following:
1. density-approximating sources (e.g., generating samples from a GMM, CNF based on training data)  hurt due to missed low-density modes ("mode discrepancy"), increasingly so the stronger/tighter the fit of the density approximation is, due to missed,  low mass modes.
2.  tightly aligning the directions of source data with training data can induce “path entanglement” and instability unless pairing is exact;

Based on these experiments the authors propose two approaches.
1. Use plain Gaussian for training, which works omnidirectional coverage.
2. Augment this with  Norm-Alignment:  which scales target norms during training to match the source (and rescales back at the end).
3. Pruning at inference: i.e. using rejection sampling to remove Gaussian samples in data-sparse  directions.  This improves FID and sampling robustness without retraining. Empirical support spans the synthetic data plus CIFAR-10 and ImageNet64.

Strengths:
The paper is largely empirical and exploratory, but does provide useful and pragmatic suggestions around source distribution construction, making it a valuable contribution.    The datasets considered are not expansive - one synthetic dataset and 2 image datasets, but the synthetic dataset and the way its constructed do elucidate the various mechanisms involved.

The mechanism of entanglement is interesting identification of a failure more for source distribution selection, for which the authors derive a informal analysis of the lipschitz constant of the flow-field and its dependence on cap-geometry constants and concentration.

Weaknesses:
1. While the paper does demonstrate efficacy of these approaches on real data, I do feel that the paper could better address other challenging types of data geometries.  For highly multimodal distributions of real data, one cannot magically expect to find an origin with respect to which the data spreads out in angular modes.   One would probably also expect significant radial multi-modality,  where there are multiple clusters of data with the same angle but different radii.  An extreme case would be concentric rings of data-points, radiating out from the origin.     It is unclear whether these approaches would be effective in that setting, or if other mitigations are necessary.
2. It would have been nice to see some other datasets beyond the CIFAR, Imagenet ones,  maybe a latent model or a conditional model, which are the most challenging settings for FM - appreciate that its addressed in the limitations

**Audience:**

Yes

**Audience Explanation:**

This paper contributes to the wider growing area of flow-matching, and would be of interest to those developing flow-matching models for their own applications, or interesting in developing novel approaches to flow-matching.

**Claims And Evidence:**

Yes

**Claims Explanation:**

Mostly yes.

The main claims are:

1. Gaussian->Pruned improves FID without retraining.
2. Norm Alignment helps (mostly at higher NFE) and can hurt at very low NFE.
3. Density approximation can hurt. DCT (mild) slightly helps; stronger GMM/CNF approximations degrade markedly.
4. Directional alignment: tight vMF + minibatch OT underperforms; oracle/global pairing shines - the authors identify path entanglement as the failre mode
5. 2D $\chi$-sphere simulation backs the story with experiments.  The appendix reports mean±std over 10 runs across multiple metrics, and the construction is explicitly derived.

**Requested Changes:**

- Figure 2 is quite hard to interpret - I would expect that as iter increases, that the source points get closer to the data, but I don't see this.
- Pruned Sampling: You define $\tau$, $\tau_r$  and the PCA test, but don’t report acceptance rates or typical resamples per kept sample. It would be useful to include both and a note on compute overhead. e.g. What % of candidates pass for CIFAR-10/ImageNet64 at your reported parameter values.  Maybe adding a small appendix plot: FID vs retained fraction (sweeping over $\tau$, $\tau_r$​), and one varying PCA dimensionality $k$ if you use a truncated basis.
-
- More practically, how do you choose these hyperparameters $\tau$, $\tau_r$, $k$, etc.

* The Tables with the image experiments list single numbers; it would be nice if you could report report mean $\pm$ std over multiple seeds, as is done for the 2D sim in Figure 7.  This would dispel the objection that some of the FID changes could be run noise.

* Are you able to explain why alignment hurts at very low NFE?

* In ImageNet64 you say "Euler integration"—is CIFAR-10 also Euler for the OT-CFM runs?

* Is the radial multimodality example described above an issue for your current approach, and if so, how can it be addressed?

* DCT: the weak/strong outcomes seem buried in the appendix - might be worth bringing them out.


Minor issues:

D1. Practive -> Practice
D1. appled -> applied.

---

> ### Author Response · Authors · 2025-08-27
> **Response to Reviewer 7yfb: Part 1**
>
> We sincerely thank the reviewer for your valuable time and constructive feedback. We have carefully considered all the comments, which have been very helpful in improving our manuscript. Below, we address each of the points raised.
>
> __[Weaknesses 1] Data Geometry Concerns__
>
> We agree that our method would have limitations in the extreme cases of data geometry you mentioned, such as concentric rings. It is for this reason that we narrow down the scope of this paper to the image distributions, which we believe are least affected by such issues. The title of our paper also reflects this scope.
>
> Nevertheless, we would like to offer a few additional points for consideration.
>
> While such geometries are theoretically possible, we claim that standard benchmark datasets (e.g., CIFAR-10, ImageNet, LibriSpeech, and common text corpora) rarely exhibit radial multi-modality.
> Moreover, when data points share the same direction but differ only in magnitude, there is a close semantic relationship between them. For example, in the case of images, data points with identical directions but different scalar values would typically represent the same visual content with only variations in color intensity or brightness. Such differences can be readily addressed through simple post-processing operations after generation, making this a non-critical issue for practical applications.
>
> __[Weaknesses 2] Limitations in Dataset Scope__
>
> We appreciate the reviewer’s suggestion to consider more challenging settings such as latent models or conditional models.
>
> - Applying to latent model
>
> Applying our method to latent models is a promising direction for future work. Our preliminary results indicate that this extension is non-trivial due to the unique structure of the latent space. Given these added complexities, we are working on latent models as an extension of this work.
>
> - Applying to conditional model
>
> We believe that our approach can be naturally extended to the conditional models as well.
> In the case of conditional distributions, the only change is that $p(x)$ becomes $p(x|c)$, and therefore all methods can be applied in exactly the same way as in the unconditional setting. For example, one can use PCA to identify directions in which the data under $p(x|c)$ are sparse or missing, and reject samples along those axes. In fact, because conditioning on $c$ typically makes $p(x|c)$ sparser than $p(x)$, this procedure can be even more effective in the conditional case.
> We will include this discussion on this in the revision.
>
> __[Requested Changes 1] Clarification on Figure 2__
>
> We thank the reviewer for pointing this out. As described in Section 3.2, we scale the approximated source so that its average norm matches the average radius of the 𝜒-Sphere. This normalization ensures that all source types start from a comparable average norm, allowing us to focus on visualizing the transport path patterns relative to the 𝜒-Sphere. While this norm scaling may reduce the apparent convergence of the source to the data, it provides a clearer and more consistent visualization of the transport dynamics across different source settings. For a visual comparison, we provide a link below to the results without this scaling, which highlights the difficulty in interpreting the transport dynamics when the source distributions are not normalized. We will include a more detailed explanation in the revision.
>
> https://docs.google.com/presentation/d/1hqL6hHTo-kA1S3IQCqRaMyHCvE5lHAtAOBeunj3ncTQ/edit?usp=sharing
>
> __[Requested Changes 2, 3] On Pruned Sampling Efficiency and Hyperparameter Choices__
>
> As you suggested, these additional experiments would indeed help readers better understand our paper. The table below reports FID scores, total generation time (for 512 images), per-image generation time, and rejection rates across different values of $\tau\_r$. The hyperparameters presented in our paper have been empirically chosen. We do not include experiments varying PCA dimensionality because PCA is used solely to identify the principal components corresponding to directions with no data variance, rather than for dimensionality reduction.
> | \$\tau\_r\$    | FID (ICFM / OTCFM)| Total Time (s) | 1 Image (ms) | Rejection Rate |
> | -------------- | ------------------| -------------- | ------------ | -------------- |
> | 0.05           | 4.00 / 4.10       | 19.96          | 38.98        | 99.94%         |
> | 0.055          | 4.05 / 4.10       | 18.11          | 35.37        | 94.90%         |
> | 0.06           | 4.13 / 4.21       | 18.09          | 35.32        | 70.08%         |
> | 0.07           | 4.22 / 4.28       | 18.09          | 35.32        | 11.88%         |
> | 0.08           | 4.24 / 4.33       | 18.09          | 35.32        | 0.97%          |
> | 1.0 (Gaussian) | 4.36 / 4.40       | 18.09          | 35.32        | 0%             |

---

> ### Author Response · Authors · 2025-08-27
> **Response to Reviewer 7yfb: Part2**
>
> __[Requested Changes 4] Statistical Significance of Results__
>
> We thank the reviewer for this suggestion. For our main experiments, we rerun the experiments and revise the results to report both the mean and standard deviation.
>
> | Train Method | Source | NFE 5 | NFE 10 | NFE 20 | NFE 40 | NFE 100 |
> | :--- | :--- | :--- | :--- | :--- | :--- | :--- |
> | OTCFM (baseline) | Gaussian | 19.80 ± 0.39 | 10.95 ± 0.34 | 7.41 ± 0.20 | 5.61 ± 0.10 | 4.40 ± 0.01 |
> | OTCFM | Pruned | **17.24 ± 0.08** | **9.17 ± 0.14** | 6.34 ± 0.05 | 4.92 ± 0.11 | 4.15 ± 0.07 |
> | OTCFM + Normshift | Gaussian | 24.92 ± 0.17 | 11.39 ± 0.01 | 6.81 ± 0.07 | 4.98 ± 0.06 | 4.03 ± 0.08 |
> | OTCFM + Normshift | Pruned | 21.52 ± 0.22 | 9.55 ± 0.06 | **5.85 ± 0.03** | **4.53 ± 0.06** | **3.88 ± 0.03** |
> | ICFM (baseline) | Gaussian | 34.49 ± 0.24 | 13.30 ± 0.06 | 7.75 ± 0.08 | 5.63 ± 0.06 | 4.35 ± 0.02 |
> | ICFM | Pruned | **28.78 ± 0.19** | **10.37 ± 0.15** | **6.40 ± 0.06** | 4.89 ± 0.09 | 3.97 ± 0.03 |
> | ICFM + Normshift | Gaussian | 52.20 ± 0.09 | 18.94 ± 0.18 | 8.10 ± 0.04 | 4.93 ± 0.08 | 3.79 ± 0.02 |
> | ICFM + Normshift | Pruned | 48.10 ± 0.34 | 16.58 ± 0.24 | 7.04 ± 0.06 | **4.56 ± 0.03** | **3.64 ± 0.02** |
>
> __[Requested Changes 5] Why Alignment Hurts at Low NFE?__
>
> This issue stems from the fundamental change in transport geometry when both source and target distributions are constrained to similar norm hyperspheres. Under a standard flow matching without norm alignment, trajectories typically follow relatively straight paths from the outer Gaussian source toward the inner data manifold. However, when norm alignment is applied, both distributions exist on similar radii, fundamentally altering the transport dynamics.
>
> As the transport operates within a batch with a finite number of samples, samples often tend to pair with targets from a region with higher density, even with optimal transport pairing. This causes the learned vector field to curve slightly toward other data-concentrated regions rather than following perfectly straight paths to the nearest points. When trajectories span from large to small radii (standard case), this pairing bias has minimal impact on path straightness. However, when both source and target exist on similar hyperspheres (when norms are aligned), the same pairing bias results in curved trajectories as the vector field gets "pulled" laterally toward high-density regions rather than following direct radial paths.
>
> This curvature becomes problematic at very low NFE, because ODE solvers require more integration steps to accurately approximate curved paths compared to straight ones. With insufficient steps, the solver's linear approximations between steps fail to capture the trajectory curvature, leading to accumulation of integration errors and degraded sample quality.
>
> To aid understanding, we provide an illustration of this phenomenon in the link below. We will include this figure in the revision to help readers see how norm alignment alters the transport geometry and why it adversely affects performance at low NFE.
>
> https://docs.google.com/presentation/d/1hqL6hHTo-kA1S3IQCqRaMyHCvE5lHAtAOBeunj3ncTQ/edit?usp=sharing
>
> __[Requested Changes 6] ODE solver of CIFAR10__
>
> We also use the Euler method for CIFAR-10, and we will make this clear in the paper.
>
> __[Requested Changes 7] Limitations in Dataset Scope__
>
> This point is closely related to our earlier discussion in the response to [Weaknesses 1]. Please refer to comment in Part 1.
>
> __[Requested Changes 8] Move DCT weak/strong Results__
>
> We agree with the reviewer that the weak/strong outcomes of DCT deserve more visibility. While we initially placed them in the appendix to avoid making the main text overly long, we will move the corresponding results into the main tables and add a concise discussion in the text to highlight the key findings.
>
> __Minor Issues__
>
> Thank you for your corrections. We will apply this to our revised version.

---

### Review · Reviewer_RrVx · 2025-08-19

**Summary Of Contributions:**

The paper discuss the source distribution for flow matching and proposes a practical frameworks that combines norm0aligned training with directionally-pruned sampling. The proposed approach facilitates stable flow learning while eliminating data sparse-region initialization during inference. Experiment on CIFAR10 demonstrates the advantages of the proposed method.

**Additional Comments:**

n/a

**Audience:**

Yes

**Audience Explanation:**

The discussion on flow matching is important.

**Claims And Evidence:**

Yes

**Claims Explanation:**

The authors provide both theoretical and practical support of their findings.

**Requested Changes:**

1. Add experiments on larger dataset.
2. Add experiments on difference tasks, e.g., image inpainting, image restoration.

---

> ### Author Response · Authors · 2025-08-27
> **Response to Reviewer RrVx**
>
> We thank the reviewer for your feedback and helpful suggestions. We address each of the comments below.
>
> __[Requested Changes 1] Experiments on Larger Dataset__
>
> Contrary to reviewer's comment in 'Summary of Contributions' that our experiments are limited to CIFAR-10, our paper also presents results on ImageNet in Table 5. We agree that experiments on even larger datasets would further strengthen the work, but we believe that the inclusion of both CIFAR-10 and ImageNet sufficiently demonstrates the effectiveness and generality of our approach.
>
> __[Requested Changes 2] Experiments on Additional Task__
>
> We agree that extending our method on different tasks such as image inpainting and image restoration would be a promising and interesting future work. However, to maintain the focus of this paper, we leave these tasks for future studies.

---

### Review · Reviewer_kdcC · 2025-09-09

**Summary Of Contributions:**

This work proposes a new 2D simulation method for analyzing the dynamics of flow matching models during training. Using the insights from the small-scale setting, they make a number of observations such as density approximation degrading performance, directional alignment suffering from path entanglement, gaussians providing more robust coverage, and norm misalignment causing poor learning dynamics. The work proses a framework to reconcile some of the observed issues such as aligning norms while directionally pruning samples. The method empirically supports their findings through experiments. They find improved generation quality with more efficient sampling.

**Audience:**

Yes

**Audience Explanation:**

The insights and analysis of the 2D learning dynamics are interesting and likely relevant to the members of the community. I am unsure whether insights from the novel 2D simulation will generally translate to the high-dimensional embedding space that typical diffusion models operate in, but it is worth investigating.

The proposed framework could be very useful if the efficiency and performance gains work for large-scale models.

**Broader Impact Concerns:**

No concerns.

**Claims And Evidence:**

Yes

**Claims Explanation:**

Yes, the experiments back the claims. I encourage the authors to perform experiments on larger-scale diffusion models as the largest dataset they benchmark on is ImageNet64. As the method doesn't require retraining, it would not be too difficult to evaluate the method on state-of-the-art models. Showing that this method works for cutting edge models would significantly increase the impact of the work.

**Requested Changes:**

I encourage the authors to perform more extensive experiments on large-scale diffusion models as I discussed above in the claims and evidence section. Though the method may work on small-scale models trained on cifar and ImageNet64, it would be nice to show that it translate to standard open-source models such as Stable Diffusion. It is not critical to acceptance, but would greatly increase the impact and usefulness of the method.

---

> ### Author Response · Authors · 2025-09-15
> **Response to Reviewer kdcC**
>
> We thank the reviewer for this insightful suggestion.
>
> First of all, we would like to clarify that our research focuses on flow matching rather than diffusion models.
>
> We agree that demonstrating scalability to large-scale models would increase our research's impact. However, most large-scale open-source models operate in latent space as conditional models, while our current work is limited to unconditional model and RGB space as mentioned in our paper's limitations, making direct application challenging. Furthermore, evaluating open-source large-scale models poses additional challenges because their dataset compositions are often undisclosed, making reliable FID measurements difficult, and without access to the training dataset it is not possible to precisely identify rejection axes.
>
> Despite these difficulties, to partially validate our method's scalability, we conducted an experiment using MeanFlow's ImageNet256 pretrained weights with a Conditional Latent Flow Model (SiT-B/2). In latent space, data-sparse directions are less prevalent than in RGB space because the KL divergence term in VAE training enforces a near N(0,I) distribution. Nevertheless, even with fewer sparse directions, our pruning method still achieved performance improvements:
> | Model             |  Source    | FID (NFE=2)|
> |-----------------|-------------|--------------|
> | MeanFlow (SiT-B/2)  | Gaussian  |     5.39         |
> |                    | Pruned    |     **5.21**         |
>
> Our current approach employs a simple PCA method for identifying data-sparse directions and does not yet employ more advanced techniques for discovering data-absent directions. Moreover, our analysis of latent space remains limited. We believe that with more sophisticated techniques for discovering sparse directions and a deeper exploration of latent structures, we could achieve greater performance improvements. Ultimately, these preliminary results demonstrate our method's potential applicability to large-scale flow matching models and provide a starting point for future research in this direction.

---

### Decision · Action_Editor_1FQE · 2025-12-01

**Recommendation:** Accept as is

**Audience:**

Yes

**Audience Explanation:**

Flow matching has recently become a major approach in generative modelling, and exploring the method wrt initial distributions is an interesting research question. All reviewers agree.

**Claims And Evidence:**

Yes

**Claims Explanation:**

All reviewers agree, although one reviewer would have liked to see more comprehensive empirical evaluation. The paper presents several empirical insights that support their claims.